# The synergistic effects of microcredit access and agricultural technology adoption on maize farmer's income in Kenya

Shadrack Kipkogei[1], Jiqin Han[1]*, Gershom Mwalupaso[1], John Tanui[2], Robert Brenya[3]

**1** College of Economics and Management, Nanjing Agricultural University, Jiangsu, P.R China, **2** Moi University, Eldoret, Kenya, **3** Commercialization Division, CSIR-Soil Research Institute, Kumasi, Ghana

* jhan@njau.edu.cn

**Data Availability Statement:** Data will be provided under the link and also excel,and stata dta file.

**Funding:** This study was financially supported by Nanjing Agricultural University in the form of a

## Abstract

Addressing global food security demands urgent improvement in agricultural productivity, particularly in developing economies where market imperfections are perverse and resource constraints prevail. While microcredit is widely acknowledged as a tool for economic empowerment, its role in facilitating agricultural technology adoption and improving agricultural incomes remains underexplored. This study examines the synergistic effects of microcredit access and agricultural technology adoption on the incomes of maize farmers in Kenya. Using household-level data, we employ an endogenous switching regression framework to control possible endogeneity in access to microcredit. Our findings shows that microcredit access positively influences the adoption of advanced agricultural technologies. Key determinants, including marital status, use of fertilizer application, access to extension services, and cooperative membership, are identified as significant determinants of microcredit access. Notably, the Average Treatment Effect on the Treated (ATT) indicates a 40.52% increase in income among farmers who access microcredit, mainly driven by the timely adoption of high-quality seeds, improved agricultural technologies, and enhanced inputs. These results highlight microcredit's role in promoting allocative efficiency and enhancing Total Factor Productivity (TFP) within agricultural systems. Robustness checks, including propensity score matching and sensitivity analyses, corroborate these findings. The study recommends the implementation of targeted financial policies and educational initiatives meant to promote credit access, encourage savings, and enhancing financial literacy, particularly for credit-constrained households. Integrating these measures could strengthen rural financial markets and drive sustainable agricultural development across the regions.

## Introduction

As the world population is projected to reach 8.6 billion by 2030, the demand for food and agricultural production requires urgent attention [1, 2]. The adoption of modern agricultural technologies is critical for enhancing productivity, increasing farm income, and alleviating poverty [3, 4]. In Sub-Sahara Africa (SSA), particularly Kenya, the adoption of agricultural

"Philosophy and Social Science Laboratories of Jiangsu Higher Education Institutions Intelligent Laboratory for Big Food Security Governance and Policy" award received by JH. This study was also financially supported by Nanjing Agricultural University in the form a of project grant, "Education Models and Pathways of Training International Talents at Universities of Agriculture and Forestry" (2020320), received by JH. No additional external funding was received for this study.

**Competing interests:** NO authors have no competing interest.

technologies is limited despite several efforts by international policy instruments such as Input Subsidy Programs (ISPs) and National Seed Certification Programs (NSCP). Additionally, the evolving pressures of agricultural pathogens and pests, exacerbated by anthropogenic climate change, further threaten crop resilience and total factor productivity(TFP) [5]. These factors impede the progress toward achieving Sustainable Development Goals (SDGs), which aim to end poverty and hunger [6].

Globally, maize is the most critical food source, with annual production of approximately 200 million metric tons [7]. In Kenya, agriculture is the backbone of the economy, contributing about 33% of the GDP and supporting 70% of the population. Maize is the primary staple food, accounting for 65% of the average caloric intake and 28% of household expenditures. Despite its significance, Kenya faces production shortfall, with an annual maize consumption of 5 million metric tons compared to the production capacity of only 3.5 million metric tons, resulting in reliance on imports [8]. Additionally, according to [9] only 2.1 million acres out of potential arable land for maize 4.1 million acres are actively farmed, highlighting underutilization of agricultural resources and inefficient resource allocation. Given this trend, policy focus improving productivity and maximizing capacity utilization as to reduce agricultural deficits and minimize reliance on external maize supplies [10].

Modern technologies existing in Kenya for maize farmers include improved seed varieties, inorganic fertilizers, mechanization, and irrigation. However, adoption of these technologies remains low among smallholder farmers, who, based on resource constraints, rely on traditional practices [11]. Fertilizer usage, for instance, averages 20 kg/ha, far below global benchmarks in regions such as China, Brazil, India, and South Africa. This indicates a critical gap in nutrient input that hampers crop yields [9]. Similarly, mechanization in Kenya stands at just 25%, implying that most farmers still depend on manual labor in various production stages, which is less efficient and constrains productivity [9]. Moreover, only 7% of Kenya's agricultural land is irrigated, leaving most farmers to rely on unpredictable rainfall patterns [12]. This insufficient utilization of available technologies reveals the urgency of improving credit accessibility, strengthening extension services, and upgrading infrastructure to promote the adoption of modern farming technologies.

In Kenya, despite agricultural potential, smallholder farmers face several barriers to the adoption of modern farming technologies. High input costs, particularly for fertilizers, improved seeds, and mechanization tools, make these technologies inaccessible to many farmers. Moreover, financial constraints remain a significant challenge, given that about 90% of rural farmers are unable to access formal credit, restricting their investment capacity [12]. Additionally, inadequate rural infrastructure and weak agricultural extension services contribute to a lack of awareness and education on the benefits of new technologies. Furthermore, cultural preferences and risk aversion lead farmers to stick with traditional methods, fearing the risks associated with adopting unfamiliar practices. These factors collectively limit technological adoption in the agricultural sector.

To address financial constraints, undeniably, the Kenyan government, has introduced various credit policies and programs aimed at supporting smallholder farmers. These include the Agricultural Finance Corporation (AFC), established in 1963 to provide credit for agricultural production, and Kilimo Biashara, launched in 2008 to offer affordable loans to small-scale farmers. The Uwezo Fund, initiated in 2013, targets youth empowerment, women, and people with disabilities in agribusiness. Additionally, in a quest to reduce the unemployment crisis, the Enable Youth Kenya Program was launched in 2016 with funding from the African Development Bank (AfDB). Notwithstanding these initiatives, supply-side impediments, including inadequate credit sources, inappropriate loan products, and protracted processes, persist in limiting credit access, especially in rural regions. Nevertheless, relaxation of supply-side

factors, by lowering interest rates and increasing microfinance access in rural areas, have remained challenging as farmers may decline to use credit due to (i) asymmetric information, (ii) risk aversion, strict lending requirements, availability of alternative financing, (iii) cumbersome loan processes. Furthermore, non-contingency terms in credit contracts can limit microcredit demand, often leading to defaults and the confiscation of household properties.

This study aims to (1) identify credit access bottlenecks and constraints affecting smallholder farmers, focusing on both formal and informal credit sources; (2) examine the determinants of credit access considering technological combinations, specifically improved maize and inorganic fertilizers among smallholder farmers; and (3) assess the effects of microcredit on net crop income between credit-constrained and non-credit-constrained households. Utilizing endogenous switching regression to address selection bias and endogeneity following [13, 14], this study also incorporates propensity score matching and sensitivity analysis for robustness.

This study makes two significant contributions to the literature. First, previous empirical studies have explored how access to credit can enhance technology adoption [15–19], as well as yields and productivity [20–22]. In a household-level panel survey on microcredit program participation in rural Bangladeshi villages, BRAC and BRDB's RD-12 program found that credit access positively impacts household income through self-employment and entrepreneurial opportunities [23]. Other studies delved on whether microcredit access influences farm efficiency and crop insurance remain a topic of discussion Ouattara, Xueping [24]. In addition to empirical literature, other related studies have examined the indirect effects of microcredit on whether its proxy for less constrained households could mitigate consumption risks and secure market imperfections [25, 26]. Finally, the latest literature has attempted to measure microcredit and technical efficiencies in farms while separating technological changes in relation to yields and productivity [20, 27, 28]. However, these studies have limitations as they often concentrate on individual technology packages without taking into account the income differences between households with and without credit constraints.

Moreover, previous studies in Kenya have primarily examined the determinants of credit access and technological adoption, focusing on mobile banking [29–31], identifying factors such as the occupation of the household head, access to financial training, and household credit liquidity constraints. Additionally, recent empirical studies have attempted to analyze the effects of microfinance savings and borrowing on digital fintech and M-Pesa-Mshwari loans for small and medium-sized enterprises (SMEs) in Kenya, [32, 33]. However, there is a lack of studies that comprehensively evaluate the impact of credit access on technology adoption, particularly in relation to its impact on household income. To our knowledge, most existing literature primarily focuses on the direct effects of credit access on technology adoption, with only a few studies addressing the diverse impacts of credit access across various agricultural yields and regions [4, 34, 35]. This study aims to fill in that gap by using an endogenous switching regression (ESR) to look at how microcredit access affects the use of technology and income outcomes across a range of agricultural yields, while also considering the possibility of endogeneity. We also provide insights on policies that can enhance and strengthen rural financial institutional partnerships within farming communities, utilizing credit for economic empowerment. This is achieved primarily by investing in technological adoption, credit relaxations, and considering flexible repayment mechanisms.

## Materials and methods

### Description of the study area

The research was conducted in Uasin Gishu County, located in the Rift Valley, Kenya, specifically covering Moiben, Kapseret, Soy, Turbo, Ainabkoi, and Kesses sub counties, as shown in

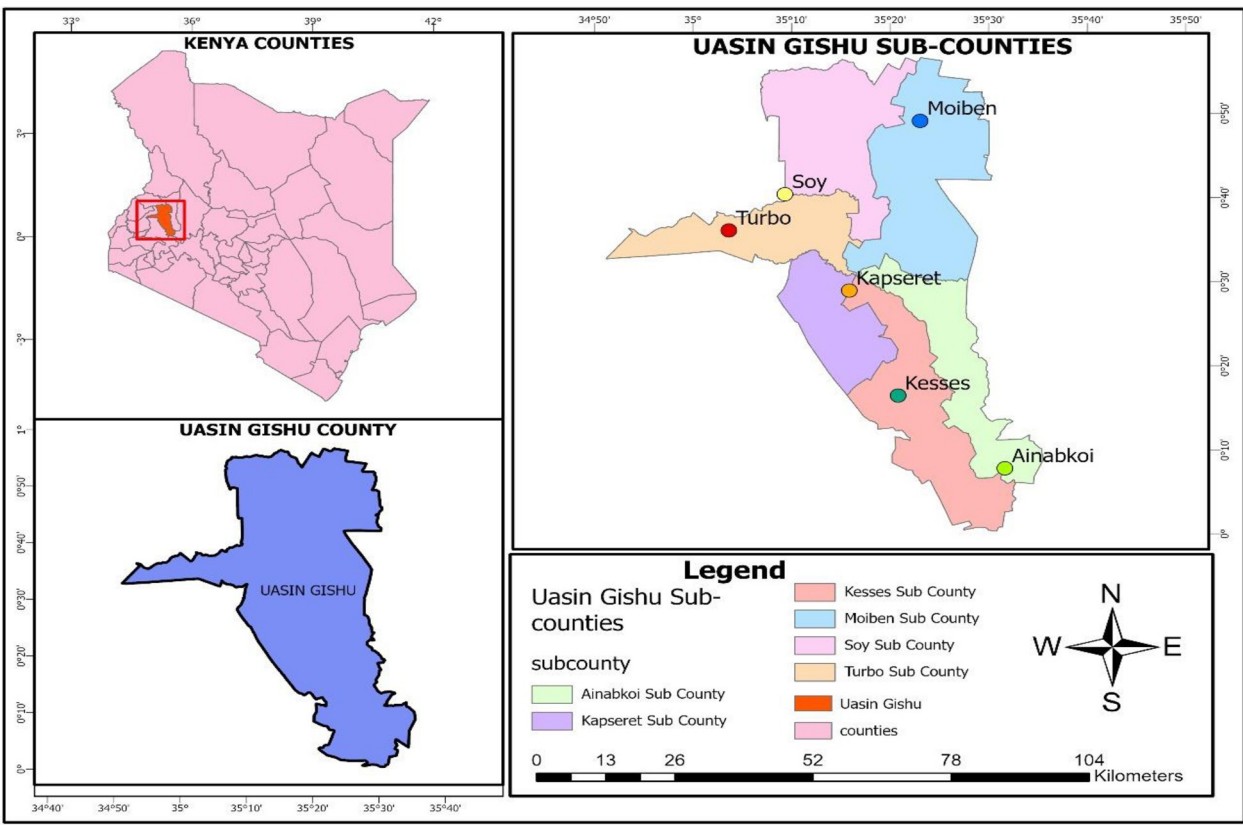

**Fig 1. Map of study areas.** Source: Author's survey.

**Fig 1**. Geographically, Uasin Gishu is situated between latitudes 0.0023˚ N and 35.3944˚ E and longitudes 0.9502˚ N and 36.4484˚ E. It borders Elgeyo Marakwet and Turkana to the north, Uganda to the west, and other key agricultural regions of Kenya to the east. The county experiences a subtropical highland climate with moderate daytime temperatures, typically ranging from 22˚C to 28˚C during the growing season. It receives an annual average rainfall of 900–1200 mm, which is conducive to maize cultivation. The region experiences its rainy season from March to October, followed by a dry period from mid-October to February. The unimodal rainfall supports rain-fed agriculture, which is the predominant method of farming in the area. Uasin Gishu's rich, fertile soils, along with its moderate temperatures and reliable rainfall, make it one of Kenya's most intensive maize-growing regions. In 2022, approximately 105,000 hectares of land were under maize cultivation, producing around 4.5 million 90-kg bags of maize [36].

This research area was chosen because agriculture is the backbone of the economy, supported by a variety of ecological zones that foster the cultivation of diverse crop types. The county is the second-largest producer of maize in Kenya, following Trans-Nzoia County, with over 70% of its farmers growing it. However, maize production has been in decline since then, dropping from approximately 42.1 million metric tons to about 36.7 million metric tons by the end of the 2022 cropping season [37]. According to the *Economic Survey* [38], this decline is attributed to factors such as financial constraints, unfavorable weather conditions, and limited access to modern agricultural inputs in maize production [39]. Apart from agriculture, residents also participate in non-agricultural enterprises, including trade, commercial transportation services, and artisanal craftsmanship. Notably, the county is globally recognized as the

"home of champions" due to its high altitudes and cool temperatures, which are ideal for training world-class athletes. Moreover, in light of technological advancements, the County's strategic potential in agricultural innovation and exhibitions has demonstrated that technology adoption enhances total factor productivity (TFP), as farmers increasingly rely on improved seeds, mechanization, and smart agriculture, reducing dependence on unreliable rain-fed agriculture and antiquated farm implements. These conditions create an ideal environment to evaluate how advancements in agricultural technology, including the utilization of enhanced seeds, mechanization, as well as accessibility to microcredit, might enhance production and income for maize farmers.

## Sampling procedures and sample size

This study utilized data from a cross-sectional household survey conducted between January 1, 2024, and March 31, 2024, targeting maize farmers. The study aimed to compare two groups: those with access to credit for adopting improved farming technologies that involve improved maize seeds, inorganic fertilizers, and pesticides, and those without such access. A two-step sampling procedure was implemented. First, study villages were selected using a probability proportionate to size sampling method, based on the 2019 census data, which reported a population of 1,163,186 with a density of approximately 368 individuals per square kilometer. In the second step, nineteen farmers per village were randomly selected from household lists provided by assistant chiefs in each village, with interviews conducted by fifteen enumerators that covered a total of 30 villages within 6 sub-counties. The sample size of 572 farmers was determined using Yamane's (1967) formula, which considers population size (N), sample size (n), and level of precision (e) [40]. This approach was chosen for cost logistics, allowing for inference to the entire population. The formula is presented as shown in Eq 1 below:

$$n = \frac{N}{1 + N(e^2)}, \quad n = 572 \tag{1}$$

This method was chosen due to its efficiency in balancing cost and logistical considerations, thereby ensuring that the findings could be generalized to the broader population. Previous studies, such as [41–43], have successfully employed Yamane's formula in similar impact evaluation contexts; hence, it proves its relevance and applicability in agricultural research. To consider the gender and variables' level of importance in the questionnaire, we implemented an Analytical Hierarchical Process (AHP) model that prioritized variables to ensure data reliability and minimize bias. The questionnaire, designed using the AHP model, decomposed the problem of credit access and technology adoption into manageable components. The collected data underwent validation checks for accuracy and consistency, and 48 households were excluded due to unrealistic outliers and discrepancies. This observation was viewed as minimizing missing data through follow-up interviews or consultations with local agricultural extension officers. Ultimately, we collected valid data from 524 farmers, dividing them into 368 treated groups and 156 control ones.

## Theoretical framework

**Pecking order theory.** Following [44], we used the pecking order theory to examine how maize farmers in Uasin Gishu financed their production process solely, without external sources, had they had access to credit facilities. Based on this theory, individuals and firms prefer to fund production inputs using internal rather than external finance. Nevertheless, when

internal funds are insufficient, firms consider external debt with equity issuance as a last resort. Based on this concept, there is a pecking order.

The Pecking Order Theory ranks financial choices considering internal and external sources. External credit is prioritized for maize farmers if more than savings and investments are needed for the current planting season. This credit can come from formal or informal sources, often depending on the farmer's relationship with lenders. Farmers generally prefer formal credit for its higher loan amounts and security despite higher interest rates. The capacity of maize farmers to access credit for technological adoption depends on interest rates, credit sources, and availability. Following [45], we model credit access as debt, as shown in Eq 2, hypothesizing that credit access positively influences technology adoption and net crop income.

$$\textbf{Credit access}, \mathbf{\Delta D_{it} = a + b_{po} DEF_{it} + e_{it}} \tag{2}$$

The symbol $\Delta D_{it}$ denotes the variation in credit availability for a particular firm at a given time t. The coefficients, *a and* $b_{po}$, show the correlations between the different variables, while *po* represents the profitability. Additionally, $DEF_{it}$ is a proxy for credit access determinants, which entails Debt history(D), earnings (E), and farmers-specific characteristics (F) at time t.

**Model estimation.**    The Analytical Hierarchical Model (AHM) is a decision-making framework to prioritize and resolve intricate issues [46]. This study's AHM analysis involves six steps, following the guidelines of [46] and the approach used by [47]: (i) Break down the complex problem into constituent factors. (ii) Develop a hierarchical structure. (iii) Construct a matrix of paired comparisons by evaluating elements. (iv) Attribute values to subjective judgments and determine the relative weights of each criterion. (v) Organize the outcomes to identify critical variables and verify the consistency of assessments. (vi) Validate the factor if the consistency ratio is $\leq 0.1$. AHP operates on three levels: Level 0 (primary objective), Level 1 (criteria procedure analysis for macro and micro factors), and Level 2 (elements related to Level 1) [48], as shown in Fig 2 below. The illustrations of this model are presented in **Fig 2** below.

**Pairwise comparison and prioritization matrix principle.**    The main focus is conceptualizing a matrix with values found in elements level two as key covariates influencing credit access decisions in the hierarchy. The opinions are then distributed, and numbers are assigned following Saaty's scale assumptions. We follow the importance of elements; element A is more crucial than B, assigned a value of 8, with B being a less important value at 1/8. Then pairwise was done to provide weights of covariates on a scale from 1 to 9, with those having less preferred element ("F" being reciprocal of value "Z" as shown in Table 1 below.

$$S_w = \frac{V_p}{V_p 1 + \cdots + V_{pn}} \mathbf{SI} = (\lambda_{\max} - n)/n - 1, \tag{3}$$

$$\mathbf{SR} = \frac{\mathbf{SI}}{\mathbf{RI}} \tag{4}$$

The prioritizing matrix for covariates is performed according to eigenvectors ($V_p$) in every criterion, as summarized in Eq 3 below.

Where sum weights $E_w$ for parameter matrix equal to 1 as illustrated in percentage, SI shows the mean vector. $\lambda$ is computed in around vector. Eq (3) further expresses the sum of each variable in the Eigenvector multiplied by its corresponding normalized weight covariate. In designing essential variables used in this study, the ratio consistency from SR should be Sr<10%, implying that the variables' results are applicable and unbiased. Nevertheless, an

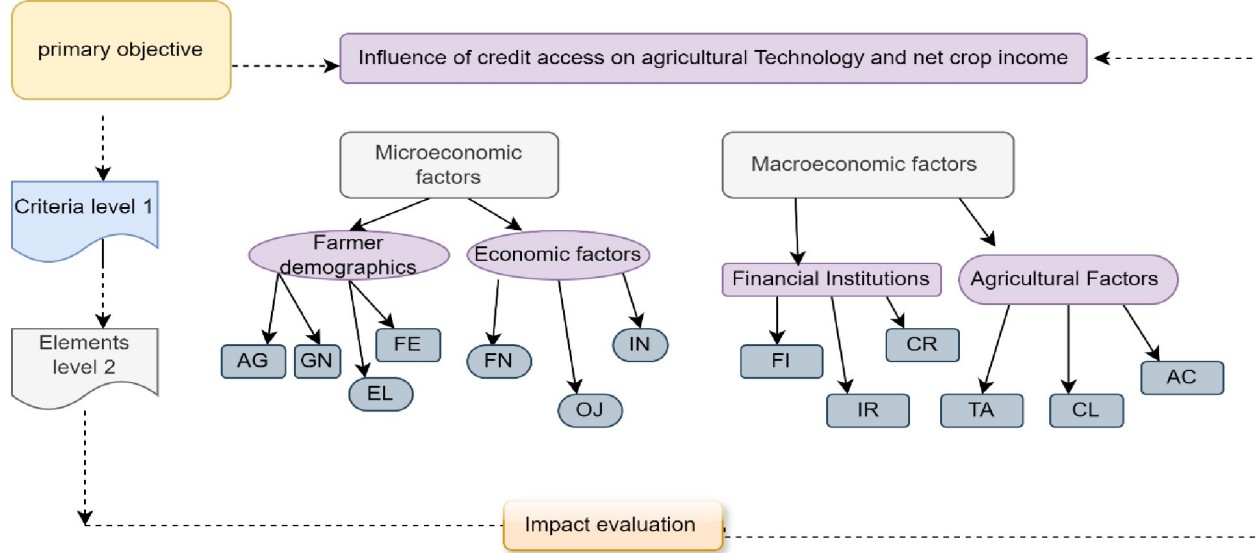

**Fig 2. AHP model of factors influencing credit access and net crop income.** An age, G gender, FE farming experience, EL education level, FN farm income, OJ of farm Income, FA farm assets, FI finance interest rates, LR loan repayments, CR collateral, TA technology adoption, CL climatic issues, AC credit access from various sources.

SR>0.1 indicates imbalanced arrangements of variable designs and requires to be revisited for balancing. Random index (RI)Estimates are illustrated in Table 2, prioritizing Agricultural factors followed by farmer demographics in the questionnaire design.

**Endogenous switching regression.**   Considering the cross-sectional data, we used propensity score and sensitivity analysis to mitigate bias from observed and unobserved confounders in assessing the treatment effects of microcredit access. Additionally, to investigate the impact of microcredit on the adoption of technology (specifically, inorganic fertilizer and improved seeds represented by dummy variables), the study utilized an endogenous switching regression (ESR). Unlike conventional methods such as instrumental variables (IV), (2SLS), or control functions strategies, it is more preferred to be flexible by separately estimating outcome equations for each regime whether participants accessed credit or not [49]. The first stage entails a probit selection equation where the dummy variable $C_i^*$ equals one if the farmer uses credit

**Table 1. Saaty's (1980) valuation scale of various elements of credit access.**

| Scale | Credit access preference variables | Considerations |
|---|---|---|
| 1 | Less important | Determinants of Credit Access Objectives |
| 3 | Moderately important | Experience and judgment |
| 5 | Important | Previous credit access on experience in favors of one variable in credit access. |
| 7 | Strongly important | One variable is strongly more important than the other in influencing access to credit |
| 9 | Extremely significant | Opinion judgment strongly favors one variable in credit access |
| 2,4,6,8 | Values for inverse preference | In case credit is not granted as essential |
| 1/3,1/5,1/7,1/9 | Inversing values for weighs | Comparison when the row element is less important than the column element |

**Table 2. Random matrix variable prioritization.**

| No of criteria | 2 | 3 | 4 | 5 | 6 | 7 | 8 | 9 | 10 | 11 | 12 |
|---|---|---|---|---|---|---|---|---|---|---|---|
| RI | 0.00 | 0.58 | 0.90 | 1.12 | 1.24 | 1.32 | 1.41 | 1.45 | 1.45 | 1.49 | 1.51 |

and 0 otherwise, as expressed in Eq 5 below.

$$C_i^* = X_{ij}\phi_1 + Z_{it}\phi_2 + \varepsilon_{it} \text{ with } C_i = \begin{cases} 1 \text{ if } C_i^* > 0 \\ 0 \text{ if } C_i^* \leq 0 \end{cases}, \tag{5}$$

Where $C_i^*$ is an unobserved latent variable of credit recipients, $X_{ij}$ represents vector control variables for households I residing in t, $Z_{it}$ represents vector instruments, and $\varepsilon_{ik}$ is a standard distribution error term with zero means and variance $\sigma_\varepsilon^2$. In the second stage, the effects of credit access on technology adoption on the outcome of interest $Y_{it}$ estimated for two regimes (credit users (R1) and non-users (R2)):

$$\mathbf{R1}: \ Y_{it}^1 = X_{ij}^1\gamma^1 + \mu_{it}^1 \text{ if } C_i = 1 \tag{6}$$

$$\mathbf{R2}: \ Y_{it}^2 = X_{ij}^2\gamma^2 + \mu_{it}^2 \text{ if } C_i = 0 \tag{7}$$

It is assumed that the error terms $\varepsilon_{ijt}$, $\mu_{it}^1$ and $\mu_{it}^2$ adhere to a trivariate normal distribution with a zero mean and a variance structure, as illustrated by Eq 8. The $\sigma_\varepsilon^2$ is the variance of the error term in the selection equation; the variances of the error terms in equations R1 and R2 are represented by the variables $\sigma_1^2$ and $\sigma_2^2$, respectively. Additionally, $\sigma_{2\varepsilon}$ signifies the covariance between εijt and μ²it, and $\sigma_{1\varepsilon}$ denotes the covariance between εijt and μ1it. Significance in either $\sigma_{1\varepsilon}$ or $\sigma_{2\varepsilon}$ would lead to the rejection of the null hypothesis that there is no selection bias.

$$cov(\varepsilon_{i1}, \varepsilon_{i2}, \sigma_{2\varepsilon}) \begin{cases} \begin{pmatrix} \sigma_\varepsilon^2 & \sigma_{2\varepsilon} & \sigma_{1\varepsilon} \\ \sigma_{1\varepsilon} & \sigma_2^2 & \sigma_{I\varepsilon} \\ \sigma_{1\varepsilon} & \sigma_{i2} & \sigma_1^2 \end{pmatrix} \end{cases} \tag{8}$$

An Endogenous regression model's primary benefit is its ability to calculate the variable's effect on treatments and counterfactuals. Following Maddala [50], this simultaneous switch leads to the ESRM model, which has been effectively examined by use of total maximum likelihood estimation in Stata as shown in Eqs 9 and 10 below:

$$EY_{it}^1|C_1 = 1) = E(v_{1F}|\varepsilon_i > -Z_{it}) = \sigma_{1\varepsilon}\left[\frac{\phi\left(\frac{Z_{it}}{\sigma}\right)}{\Phi\left(\frac{Z_i\beta}{\sigma}\right)}\right] \equiv \sigma_{1\varepsilon}\lambda_1 \tag{9}$$

$$E\left(Y_{it}^2|C_1 = 0\right) = E(v_{1I}|\varepsilon_i, \leq -Z_{it}) = \sigma_{2\varepsilon}\left[\frac{\phi\left(\frac{Z_{it}}{\sigma}\right)}{\Phi\left(\frac{Z_{it}}{\sigma}\right)}\right] \equiv \sigma_{2\varepsilon}\lambda_2 \tag{10}$$

Where $\phi$ represents the standard normal probability density function, $\Phi$ represents the standard normal cumulative density function, and $Z_{it}$ is used for instruments that are included in Eqs (6) and (7) [51]. Conditional expectations, outcomes, and counterfactuals can be computed and expressed following [52] as shown below:

$$E(Y_{it}^1 \mid C = 1) = X_{ij}^1\gamma^1 + \sigma_{1r}\lambda^1 \tag{11}$$

$$E(Y_{it}^2 \mid C = 1) = X_{ij}^1 \gamma^2 + \sigma_{2\varepsilon} \lambda^1 \tag{12}$$

$$E(Y_{it}^1 \mid C = 0) = X_{ij}^2 \gamma^1 + \sigma_{1\varepsilon} \lambda^2 \tag{13}$$

$$E(Y_{it}^2 \mid C = 0) = X_{ij}^2 \gamma^2 + \sigma_{2\varepsilon} \lambda^2 \tag{14}$$

Where $\lambda^1$ and $\lambda^2$ represent the inverse mills ratio that results from the selection equation for credit users and non-users, respectively. Eqs (11) and (14) show observable expected outcomes, Eq (12) represents counterfactual expected outcomes for noncredit users, and Eq (13) shows counterfactual expectations if noncredit users would have obtained credit access. Finally, we estimated the Average treatment effect on treated (ATT), from differences from Eqs (11) and (12), as it provides the change in average outcome from treated households (credit users), unlike treatment effects on untreated (ATU) which can be estimated from the difference in Eqs (13) and (14) as it's not considered due to tendency of giving unreliable policy implications. We therefore computed ATT as expressed in Eq (15)

$$\text{ATT} = E(Y_{it}^1 \mid C = 1) = X_{ij}^1 \gamma^1 + \sigma_{1r} \lambda^1 - E(Y_{it}^2 \mid C = 1) = X_{ij}^1 \gamma^2 + \sigma_{2\varepsilon} \lambda^1 \tag{15}$$

## Propensity score matching (PSM) for robustness check

The Propensity Score Matching (PSM) technique is applied as a robustness check to control for potential biases in the estimation of treatment effects, ensuring that the results from the ESR model are reliable. PSM helps match credit users and non-user groups with similar observable characteristics, thereby reducing selection bias and ensuring that comparisons between groups are valid. Once the propensity scores are calculated, matching methods like nearest neighbor, kernel matching, and caliper are used to pair adopters with non-adopters who have similar propensity scores. This lets us figure out how the treatment affects income and productivity, as shown in Eqs 16 and 17.

$$\text{Step 1}: \text{Participation Decision. } P(Y = 1) = \emptyset(X\beta 1 \tag{16}$$

$$P(y = 1) = \emptyset(X\beta 1) = P(Z \leq X\beta 1)$$
$$\text{ATT} = E[Y(1) - Y(0) \mid T = 1] \tag{17}$$

Where $P(y = 1)$ represents the participation decision, $\emptyset$ the cumulative distribution function (CDF) of the standard normal distribution, is mostly used in probit models, $X$ represents explanatory variables and $\beta 1$ represents the estimated coefficients for those variables. $Y(1)$ is the outcome for the treated and $Y(0)$ shows the outcome for the control group. After matching is done, the estimation of the Average Treatment Effect on the Treated (ATT) is computed by comparing the outcomes of treated and matched untreated units.

## Ethical considerations

The study was approved by the National Commission for Science, Technology, and Innovation (NACOSTI), License No: NACOSTI/P/24/34867, with approval granted prior to data collection. Due to resource constraints, verbal consent was obtained from participants and was documented using phone recording software. Local representatives, known as 'respected leaders' in each survey village, were consulted to verify the appropriateness of the consent process and were identified as the trusted voices for their communities. Participants were fully informed

about the study's objectives, procedures, potential risks, and their rights, and they were assured that participation was voluntary and confidential. The ethics committee was also informed that minors (under 18 years of age) would provide their own consent. All data collected procedures and analysis were securely stored and accessible only to the research team ensuring participants could not be identified by number or name. Date: 01/01/2024.

## Results and discussions

### Descriptive statistics

The results in Table 3 summarize the social economic characteristics of the maize farmers illustrating variables used in the model estimation. The dependent variable, farmers' level of technology adoption farmers using microcredit provides perceived insights into demographic and economic adoption patterns. The average age of maize farmers is 45 years, relatively younger than 46 years for non-users, indicating that farmers are relatively young, with productive capabilities of adopting technological advancement that could enhance productivity. Results corroborate empirical findings by Bakare, Ogunleye [53] and Gabriel and Gandorfer [54] who found that the age of 41–50 is a critical point of productivity and receptivity to innovation, young farmers possess the skills and risk tolerance to incorporate new agricultural technologies effectively. Additionally, findings correspond to findings by Asante-Addo, Mockshell [55] in Ghana who found that younger farmers exhibit a proactive stance toward technology adoption, distinctive attributes caused by familiarity with digital tools, and a heightened propensity

**Table 3. Descriptive statistics on mean comparison differences between households with credit access and non-users.**

| Variables | Description | Credit users (N = 368) | | Without Credit (N = 156) | | Difference |
|---|---|---|---|---|---|---|
| | | Mean | SD | Mean | SD | |
| Chemical fertilizer | whether household uses inorganic fertilizers (1 = yes,0 otherwise) | 0.89 | 0.02 | 0.19 | 0.03 | -0.70** |
| Improved varieties | whether household uses improved maize varieties (if yes = 1,0 = No) | 0.85 | 0.02 | 0.80 | 0.03 | -0.04 |
| Age | Age of household head in (years) | 44.93 | 0.56 | 46.35 | 0.97 | 1.41 |
| Gender | Sex of the household head (1 = Male,0 otherwise) | 0.38 | 0.03 | 0.28 | 0.03 | 0.03* |
| Marital status | Marital status of the households (1 = married or otherwise) | 0.71 | 0.02 | 0.68 | 0.04 | -0.03** |
| Occupation of household | Household's head occupation (1 = farming,0 = other business) | 0.60 | 0.03 | 0.65 | 0.04 | 0.65 |
| Education level | Education level of household head in (1 = educated or otherwise) | 12.87 | 0.36 | 10.30 | 0.43 | 2.56 |
| livestock ownership | Household ownership of livestock (1 = yes, 0 otherwise) | 0.44 | 0.02 | 0.39 | .039 | -0.05* |
| Land size in (ha) | Size of the land occupied by the household in acres | 3.72 | 0.12 | 2.41 | 1.38 | -1.30* |
| Extension Education | Household head access to extension (yes = 1) | 0.74 | 0.02 | 0.44 | 0.03 | -0.44*** |
| Access to microcredit | If the household received credit for (1 = yes,0 = No) | 0.88 | 0.02 | 0.20 | 0.03 | -0.68*** |
| Sources of information | Sources of information (1 = media,0 = otherwise) | 0.52 | 0.04 | 0.22 | 0.02 | 0.31 |
| Cooperative membership | Household head group membership (1 = member,0 = No) | 0.79 | 0.02 | 0.29 | 0.04 | -0.49*** |
| Nonfarm income | whether households depend on farming only (if yes = 1,0 = No) | 1.79 | 0.021 | 1.79 | 0.03 | -0.01 |
| Farming experience | Number of years in farming (1 = more than ten years,0 below 10) | 0.78 | 0.02 | 0.75 | 0.03 | -0.04 |
| Bank account | Household membership in a financial institution,1 = yes,0 otherwise) | 0.89 | 0.02 | 0.52 | 0.04 | -0.37*** |
| Social interactions | Whether the household head participates in community social groups | 1.93 | .012 | 1.87 | .026 | -0.06 |
| Distance to finance institutions | Distance from village to Agricultural banks in (km) | 1.98 | 0.04 | 2.00 | 0.08 | 0.01 |

Notes

*** p < .01

** p < .05

* p < .1 Ksh represents Kenya Shillings, which was equivalent to 1USD = 132 Ksh in Survey (2024) Source: Author's processing

for risk-taking. In contrast, as farmers age, the adoption likelihood tends to diminish, as older farmers often prioritize stability over potential yield-enhancing investments due to increased conservatism and reliance on accumulated savings rather than external financing [56]. This pattern resonates with microeconomic theories of diminishing marginal utility in risk aversion, where the marginal benefit of new technology does not adequately outweigh the perceived risks among older farmers, Sunding and Zilberman [57].

Additionally, most maize farmers about (65%) are men. This implies that maize farming is a male-oriented occupation. This is plausible because of the socio-cultural norms in African agricultural systems, where men traditionally possess greater mobility, community participation, and decision-making authority [58]. Consequently, male farmers have better access to technological and information sources approximately (78.88%) as compared to nonusers (47.42%), as well as productive assets endowments such as land. In practice land ownership in Kenya is a significant factor in technology adoption, this is evidenced by economic assets facilitating credit access by serving as collateral [59]. Consequently, with the existence of cultural inheritance patterns, land ownership still gives men to dominate, limiting female farmers' access to credit and their ability to adopt capital-intensive technologies [60].

The majority (71%) of the credit maize farmers are married. This aligns with the findings of [61], which postulated that married people mostly engage in farming activities. The likelihood of adoption of technology adoption has been found to diminish among unmarried maize farmers [62]. These implications could be attributed to a lack of access to productive resources and assets that can be available through marriage including land ownership as these social characteristics drive farmers to seek microcredits to enhance technical efficiency [63, 64]. In other comparisons, findings showed that credit-accessible farmers owned livestock, were more educated, received extensions, belonged to cooperatives, owned bank accounts, and were more experienced. This corroborates the studies by [65, 66] that possession of important assets and skills promotes technology adoption. Further, both groups have adopted over 80% improved maize seed varieties, but credit users show higher adoption rates of inorganic fertilizers (89%) in comparison to noncredit users (19%) implying that access to these inputs is seemingly relatively expensive though crucial for optimal yields. Noncredit users generally have fewer than ten years of schooling, less involvement in financial institutions (52% have bank accounts versus 89% for credit users), and lower access to extension education and information. Distance to financial institutions and markets varies significantly. Credit users have shorter distances to markets (0.16 km) and financial institutions (1.98 km), whereas noncredit users face longer distances to markets (0.91 km) and financial institutions (2.00 km). Given that Table 3 does not provide differences in determinants and income impacts of credit access, a more robust approach is necessary; otherwise, an inexperienced estimator is likely to either under/or overstate the findings. Further, we tested for multicollinearity and heteroscedasticity as illustrated **S1 Table.** According to Gujarati [67], VIF more than the threshold value of 10 indicates multicollinearity within covariates. Based on our results, as indicated **S1 Table** we found (VIF) of 1.27. Additionally, results from the Breusch-Pagan (BP) test on heteroscedasticity test results show a Chi-square value of 0.025, with a p-value of 0.97. This high above p-value leads us to fail to reject the null hypothesis, confirming constant error variance and no evidence of heteroscedasticity [68].

## Access to credit use and different requirements by maize farmers

Table 4 illustrates sources of credits within the reach of maize farmers in Uasin Gishu County. Results indicate that approximately 70.23 households had access to credit during the previous planting season. Usually, farmers in most rural areas obtain credit from various sources,

**Table 4. Microcredit sources of maize farmers in Uasin Gishu.**

| Sources of microcredit | Frequency | Percentage |
|---|---|---|
| Agricultural Finance Cooperation -Government programs | 134 | 34.7 |
| Apollo Agriculture | 68 | 17.6 |
| Funds from other enterprise /personal finance | 156 | 100 |
| One Acre Fund -Social Enterprise | 47 | 49.7 |
| Saccos and Cooperative's | 36 | 20.80 |
| Family and Friends | 27 | 7 |
| NGOs | 0 | 0 |
| **Agricultural technology adoption** | **Frequency** | **Percentage** |
| Purchase of improved seeds | 272 | 69.4 |
| Purchase of inorganic fertilizers | 341 | 87 |
| Pesticides and agrochemicals | 213 | 54.6 |
| Equipment and machinery | 70 | 17.9 |
| Renting agricultural land | 150 | 29.1 |
| Other business activities | 127 | 24.7 |

Source: Author's processing

including banks, non-government organizations, Saccos, farming groups, government agencies, family, and relatives [53, 69]. Our study findings reveal that 7% of maize farmers seek credit from family and friends and 17.6% from Apollo Agriculture. In addition, findings indicate that 20.80% obtain credit from saccos and cooperatives, 34.7% from Agricultural Finance Cooperation, and 49.7% from one-acre funds. The two nonprofit social organizations (Apollo and One Acre Fund) are depended upon by most rural farmers in Uasin Gishu County as they provide microloans, inputs, and agricultural training.

In contrast, reported data indicates that credit from formal sources had more restrictions, particularly on credit limits and collaterals; poor farmers prefer other alternatives, as mentioned in previous sources. Further, results show that most maize farmers operating other business enterprises (100%) depend on their funding from various business engagements from past cropping seasons. Access to credit services is essential for farmers to adopt agricultural technologies Makate, Makate (35). Therefore, the lack of sufficient access to credit could limit farmers' ability to implement new agricultural technologies in maize farms, as they may require assistance to cover the costs of necessary inputs and optimize the utilization of existing technological information [70]. Table 4 also presents the purpose, primary intentional usage, and target of the farmers. Farmers usually use microcredit to acquire inputs such as fertilizer, machinery, improved seeds, land preparations, and harvesting and threshing labor. Based on According to the study findings, 87% of maize farmers use credit to purchase inorganic chemical fertilizers, 69.4% to buy improved yield seeds, pesticides, and agrochemicals, 54.6% to rent agricultural land and land preparations, 29.1% to finance other business ventures, and 17.9% to buy farming equipment and Liverpool-Tasie, Omonona [71] also corroborate the study's findings, indicating that maize farmers allocate their credit towards the purchase of improved seeds and inorganic fertilizers.

## Self-reported impacts of low agricultural technology adoption on maize yield

The variation in agricultural production caused by climate variability and extreme weather events negatively influences cereals, legumes, and tubers, resulting in low yields. Based on

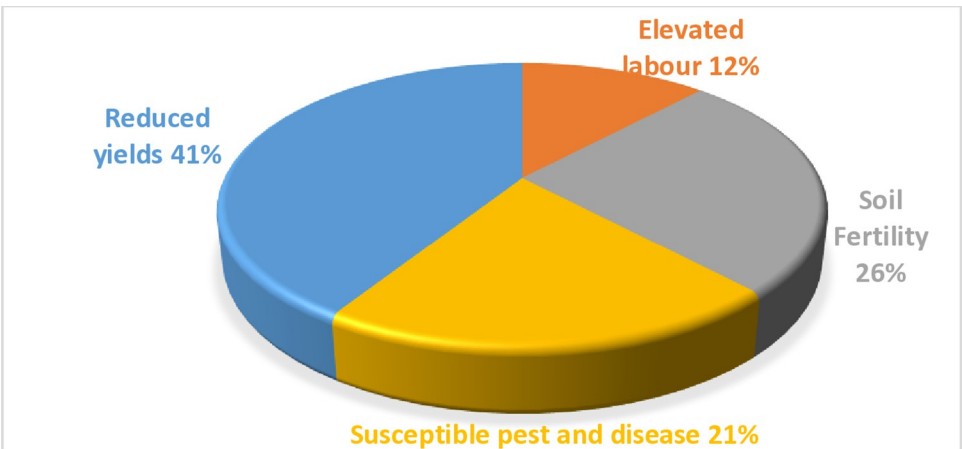

**Fig 3. Self-reported impacts on causes of low adoption of agricultural technologies on maize yields.** Source: Author's processing.

study results, as shown in **Fig 3**, approximately 97% of maize farmers responded that climate variability impacted maize farms during the previous cropping season. This was attributed to a delay in rainfall, which had been anticipated to begin in early April but was witnessed in mid-month of May. Furthermore, 12% of maize farmers responded that elevated labor expenses, 26% reported soil degradation as they need to apply more inputs, 21% reported higher susceptibility to pests and disease, and 41% reported reduced maize yields. Adopting agricultural technology has been considered a critical element that can influence farmers with credit access to fully adapt to perceived climate changes through implementing climate-smart agriculture and using superior farm inputs that guarantee technical efficiency and improve productivity [72].

## Discussion

### Determinants of microcredit access and effects on household income

The results in Table 5 are derived from an endogenous switching regression (ESR) model, estimated using Full Information Maximum Likelihood (FIML), to assess the determinants and impact of credit access on maize farmers' household income. The lower part of Table 5 shows that coefficients for $\rho_1$ and $\rho_2$ are not statistically significant, indicating that there is no significant correlation between the unobserved confounders influencing income and the decision to access credit. The likelihood ratio test, conducted at the 1% level, confirms the joint dependence of the three equations, further supporting the validity of the model. The variation in coefficient estimates for credit users and non-users highlights the superiority of the switching regression model over a simple treatment effect model [73]. Notably, variables such as sources of information, land size, and the ability to use chemical fertilizers, are key essentials in complementing improved maize varieties in the study area. These results reveals that there is unobserved heterogeneity that could cause bias if it is not taken into account. This is why the ESR model is the best choice for this analysis.

To ensure proper identification in the ESR model, [74] proposed that the criterion function must integrate all explanatory variables from the regime equations, along with instrumental variables. We included the distance to financial institutions in this regression. Whereas this selected instrument correlates with individual adoption behavior, it was uncorrelated with maize yield income, thereby meeting the exclusion requirement necessary for valid

**Table 5.  Estimation of the endogenous switching regression model using FIML.**

| Explanatory Variables | Access to credit | | Credit users (Regime1) | | Noncredit users (Regime2) | |
|---|---|---|---|---|---|---|
| | (1) | (2) | (3) | (4) | (5) | (6) |
| | Coeff | St. Err | Coeff | St. Err | Coeff | St. Err |
| Age | 0.003 | (0.008) | 0.001 | (0.004) | 0.003 | (0.005) |
| Gender | -0.127 | (0.203) | -0.148 | (0.090) | -0.004 | (0.161) |
| Marital Status | 0.497** | (0.206) | 0.147 | (0.098) | 0.348** | (0.168) |
| Education level | 0.134 | (0.197) | 0.039 | (0.093) | 0.370** | (0.176) |
| Livestock ownership | 0.145 | (0.190) | 0.201** | (0.089) | -0.067 | (0.145) |
| Land size(ha) | 0.073 | 0.077 | 0.130*** | (0.037) | 0.073 | (0.055) |
| Occupation of household | -0.206 | (0.191) | 0.094 | (0.090) | 0.234 | (0.145) |
| Use of inorganic fertilizer | 1.708*** | (0.193) | 0.404** | (0.170) | 0.084 | (0.301) |
| Land-ownership | 0.0488 | (0.018) | -0.032* | (0.018) | 0.003 | (0.054) |
| Social interactions | -0.0220 | (0.294) | -0.071 | (0.169) | 0.046 | (0.214) |
| Farming Experience | 0.187 | (0.210) | -0.118 | (0.102) | 0.002 | (0.148) |
| Cooperative membership | 0.543** | (0.218) | 0.375*** | (0.126) | -0.447** | (0.181) |
| Source of information | 0.152 | (0.216) | -0.372*** | (0.118) | -0.308** | (0.151) |
| Extension education | 0.583*** | (0.219) | 0.376*** | (0.118) | 0.541*** | (0.163) |
| Non-farm income | 0.408 | (0.527) | -0.257 | (0.324) | -0.163 | (0.310) |
| Distance to finance services | 1.332*** | (0.202) | | | | |
| Constant | 5.071*** | (1.363) | 11.95*** | (0.832) | 8.159*** | (0.833) |
| Model Diagnostic | | | | | | |
| sigma_1/ sigma_2 | | | -0.255*** | (0.038) | -0.218*** | (0.068) |
| rho_1/ rho_2 | | | 0.271 | (0.233) | 0.476 | (0.309) |
| N | 524 | | 368 | | 156 | |
| LR test of Index. Eqns = X$^2$ chi2(1) | = 3.75*** | Prob > | chi2 = 0.052 | | | |

Notes

*** p < .01

** p < .05

* p < .1. In Regime 1 and Regime 2, the dependent variable is income for farmers using credit and non-users, respectively

Source: Author's compilation

instrumentation [75]. The results presented by ESR are presented in three sections, as indicated in Table 5. Column (1) reveals the selection equation, representing the factors influencing farmers' credit access decisions. As per regressions, important positive and significant variables were observed in social economic variables such as marital status, use of inorganic fertilizer, cooperative membership, access to extension education, and distance to financial institutions.

Given that lenders typically assess ownership when making lending decisions, marital status can influence credit access based on the ownership of family assets, reflecting the socioeconomic stability of the household. This finding corroborates [76–78] study results that married farmers may be more likely to access credit due to land ownership and other family structures. Consequently, this builds higher levels of trustworthiness among financial lenders. The use of inorganic fertilizer influences credit access decisions by enhancing the perceived productivity of this agricultural technology, which not only improves yields but also ensures sustainable income [79]. Additionally, the use of inorganic fertilizers with other inputs, such as improved seeds, signals to the financial lenders that farmers commit to improving technical productivity, thus increasing a farmer's credit solvency. The findings of this study resonate with the concept

of product-factor productivity, [80] as it suggests that the efficient allocation of inorganic fertilizers could significantly boost crop yield per unit of input, thereby enhancing a farmer's overall income stability. By increasing productivity, fertilizers guarantee higher returns from their investments, which in turn makes them more attractive to lenders, as most farmers may not have the financial means to access credit. The findings corroborate the findings [81] that access to credit is essential in technology adoption because the cost involved cannot be easily met. In contrast, studies by Wu, Hao [82] found that in some instances, inorganic fertilizer in terms of factor-factor productivity influences relationships with other factors of production such as labor and capital, creating high upfront costs and financial risks when fertilizer is overused. Access to extension education gives farmers access to credit, enhancing their knowledge and skills in areas such as crop choice, input selection across various technologies, farm planning, with a focus on improving agricultural practices, financial literacy, and understanding credit terms. This implies that credit access increases farmers' financial literacy and enhances their ability to utilize credit effectively. The study's findings align with the findings of studies [83, 84] which suggest that extension education significantly facilitates the transfer of agricultural technology among rural farm settlements through demonstrations and focus groups.

Finally, cooperative membership and distance to financial services are positive and statistically significant in influencing access to credit. Distance to financial institutions, particularly for rural farmers, is a barrier to effectively assessing credit, as time and transactional costs require transportation costs, frequent assessment of information, and more direct contact interactions with lenders [85]. Regarding cooperative membership, it influences credit access by providing farmers with collective bargaining power, facilitating the pooling of resources. Additionally, cooperatives act as a bridge between farmers and financial institutions, consequently improving trust and reducing transaction costs. The study findings align with those of studies [86, 87], which suggest that cooperatives enable the optimal use of inputs, leading to increased technical efficiency. On the other hand, a study conducted by [88, 89] revealed that poorly managed cooperatives may be unable to provide access to credit, potentially leading to a decrease in the trust and willingness of farmers to effectively advocate for their interests.

As shown in Table 5, columns (3) and (5) show estimates for the second phase in the switching regression model for how access to credit affects the household income of adopters and non-adopters. This corroborates ESR effectiveness other than just regressing data in the same model. For instance, livestock ownership positively impacts household income, as it has the potential of an additional income source and serves as a financial buffer against agricultural income volatility [90]. Owning livestock offers diversified income streams and resilience against crop failure, enhancing overall economic stability for farming households. The study findings by Nikam, Ashok [91], which suggest that livestock ownership impacts yield, are mitigated by the ease with which farmers can access credit due to the assurance of collateral. Larger land size and ownership are positively associated with increased household income, in terms of economies of scale and the potential for higher crop yields [92]. In these findings, it is expected that since total factor productivity and allocative efficiency depend on productivity per unit area, an increase in per unit leads to higher yields, translating to higher income. The findings align with studies [93, 94] and reveal that larger landholdings implement advanced agricultural technologies, techniques, and inputs, which enhance their total factor productivity (TFP) and result in an average income increase of 12–20% when compared to smaller landholders.

Additionally, the land tenure system provides security, incentivizing farmers to invest in soil fertility and yield-enhancing technologies, which, in turn, contribute to higher income. The findings are supported by Ren, Liu [95], who state that increasing farm size positively impacts farmers' net profit. The use of chemical fertilizers is positively correlated with

household income and plays a role in enhancing crop yields and agricultural productivity [96]. Fertilizers increase nutrient availability, supporting crop growth and leading to higher yields, which directly boosts farmers' income. The results contradict findings from [97] that increased chemical fertilizer usage increases rural-urban income. However, considering the need for loan repayments from credit users, a study by Jayne and Rashid [98] supports this negation by stating that rates of fertilizer applications for farmers under such programs have effects that outweigh the actual benefits received. For sustainable and ecological concerns, a study proposes optimal fertilizer usage instead of overuse, by promoting the principle of returning straws to crop fields after harvesting through an innovative decomposing mechanism. The simultaneous process could reduce environmental pollution while minimizing inputs and ensuring farmers' higher yields.

## Matching algorithm and quality matching

When calculating ATT, the covariates must be balanced to ensure that treatments and control groups are comparable. In PSM, we prioritized plausible matching techniques such as nearest neighbor, kernel, and caliper matching, based on criteria such as equal means test, balancing test, pseudo-R2, and matched sample size, as per the recommendations of Vairetti, Gennaro [99]. We preferred lower pseudo-R2 values to indicate larger sample sizes and effective balancing of explanatory factors [100]. Quality matching in PSM is essential for ensuring that the treatment effect estimates are reliable and unbiased. As illustrated in **Fig 4**, the balance of covariates has been successfully achieved, thereby ensuring a region of common support. This balance helps to eliminate bias by aligning the distributions of observable characteristics between treated and control groups, as also evidenced in **Fig 5**, which shows a significant reduction in bias across these characteristics.

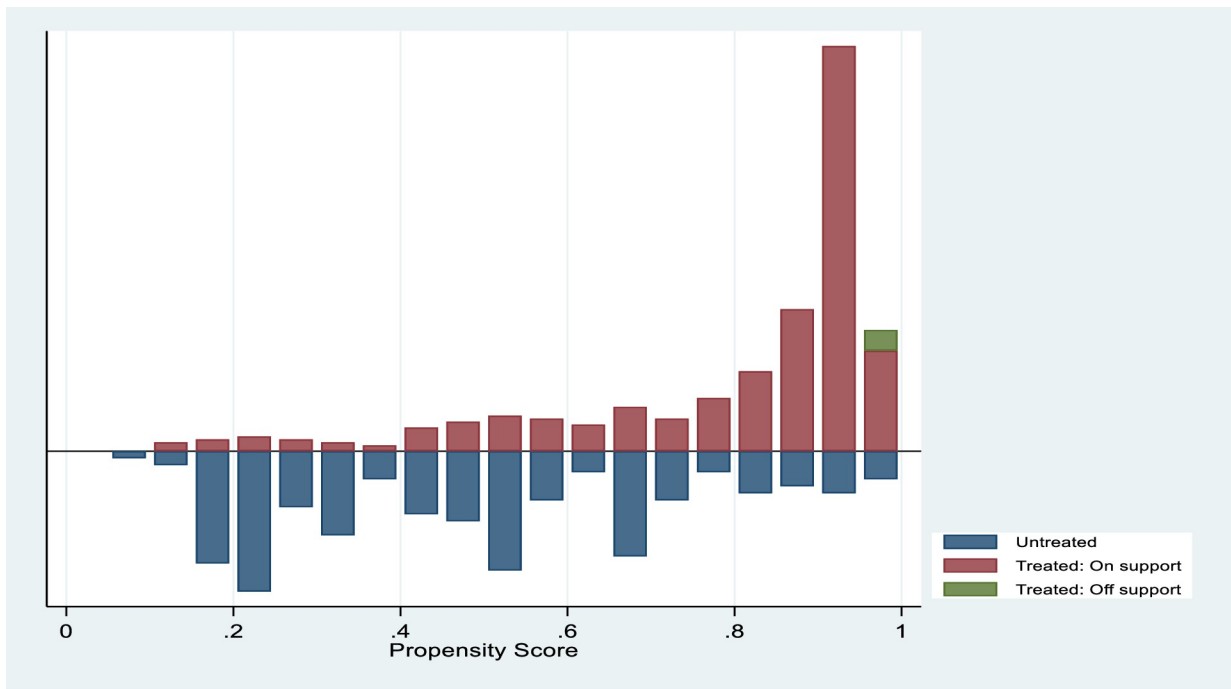

a)

**Fig 4. Balancing of covariates with 14% observations off support.** Source: Authors' computations.

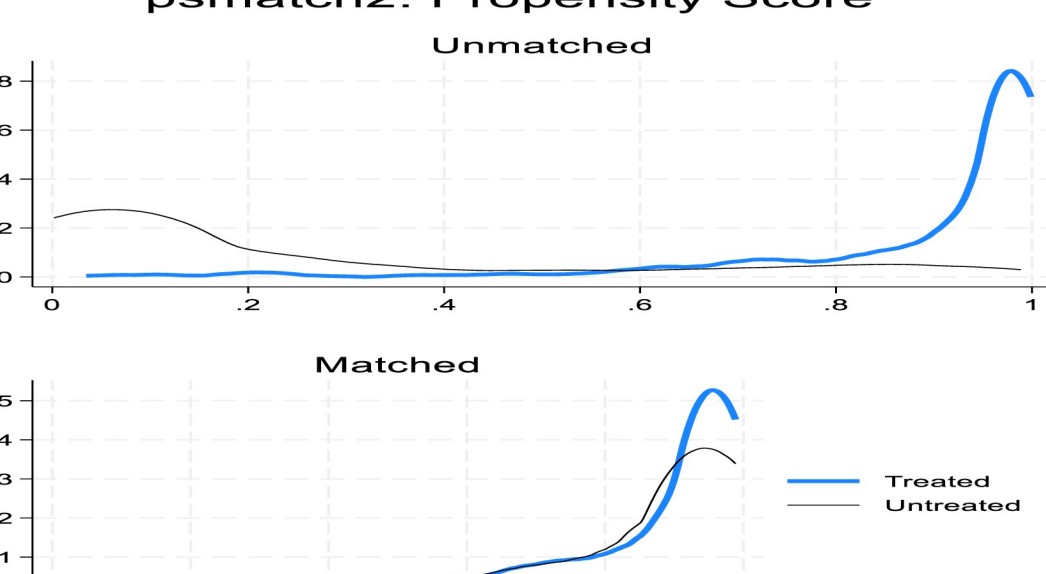

**Fig 5. Bias reduction after matching-comparable observable characteristics.**

## Impacts of microcredit access on income

Table 6, unlike the mean difference presented in Table 3, shows the impact of microcredit access on maize farmers' income, incorporating propensity score matching (PSM) and endogenous switching regression (ESR) to address and account for selection bias. The average treatment effect on the treated (ATT) is estimated by calculating the average difference in income outcomes between matched treated and non-treated individuals, Rosenbaum and Rubin [101]. To ensure robustness, various matching techniques were applied, including nearest neighbor matching (PSM-NNM), kernel matching, radius matching, and weighted nearest neighbor matching, following procedures by Caliendo and Kopeinig [102].

In the analysis, as shown in Table 7, the PSM-Neighbour (1) approach matched each treated individual with the five nearest non-treated individuals based on their propensity scores. In contrast, kernel matching applied a nonparametric technique to calculate a weighted average of outcomes, with weights inversely related to the distance between treated and control observations. Radius matching was implemented with a caliper of 0.01, limiting matches to those within a specified propensity score distance. Additionally, weighted NNM, which bases similarity on a weighted function of covariates rather than propensity scores, was applied to

**Table 6. Impacts of microcredit participation on household income from the ESR model.**

| Adoption status | N | Income from credit | Non-credit users | Treatment effect | % Change |
|---|---|---|---|---|---|
| Adopters | 368 | 12.31 | 8.76 | ATT:3.55*** (0.026) | 40.52 |
| Non-adopters | 156 | 11.68 | 8.45 | ATU: 3.23***(0.050) | 38.22 |

*Note*: Statistical significance at the 1% level. Standard errors are shown in parentheses. Credit access impact on technology adoption. ATT–Average Treatment Effect on the Treated; ATU–Average Treatment effects on untreated.

Source: Authors compilation

**Table 7. Robustness checks; propensity score matching (PSM) estimates impacts on income.**

| Matching type | Income from credit | Non-credit users | Treatment effect ATT | % Change |
|---|---|---|---|---|
| Nearest neighbor (1) | 12.29 | 8.19 | 4.08 | 49.82 |
| Kernel | 12.29 | 8.11 | 4.16 | 51.29 |
| Radius | 12.29 | 8.45 | 3.82 | 45.44 |

Note*s*

*** Statistical significance at the 1% level; ATT–Average Treatment Effect on the Treated.

validate the reliability of the findings, resonating with findings by Ricome, Barreiro-Hurle [103] The covariates used were consistent with those in endogenous switching regression.

The ESR model results show that access to microcredit significantly increases maize farmers' income by 40.52% compared to non-users. This finding is also consistent with the PSM results, which are positive and statistically significant. However, as expected, differences are observed in Tables 6 and 7, with PSM showing relatively higher estimates in comparison to ESR. The plausible explanation is that the ESR model controls for unobserved factors by estimating a selection equation, which can isolate treatment effects more precisely; in contrast, PSM may sometimes overestimate or underestimate treatment effects due to its reliance on observed covariates [104, 105]. Nevertheless, as postulated by [106] the PSM to covariate imbalances or hidden bias, estimates of causal impacts compared to models that control for unobserved characteristics, hence these variations highlight the importance of considering unobserved heterogeneity in the analysis.

Robustness checks, including t tests, confirm the significance of these findings at the 1% level. This implies that farmers with access to microcredit achieve higher incomes than non-credit users. This could be premised that the availability of microcredit facilitates the timely adoption of better-quality seeds, improved agricultural technologies, and inputs [4, 107].Additionally, microcredit access could be used to diversify income sources and set up insurance schemes to manage risks from climate shocks or market fluctuations [108]. In contrast, other studies found that microcredit access resulted in no impact on yield improvement but increased income on off-farm activities [78, 109, 110]; additionally, [111, 112] found that in general, farmers face uncertainty or poor market access and may prioritize consumption over long-term investments in productivity-enhancing technologies.

To this end, while microcredit has shown positive effects in terms of input adoption and income growth, its overall effectiveness depends on complementary factors such as education, technical support, and risk management strategies [113]. Per se, maize farmers utilized the accessed credit for the primary intended purpose, particularly the adoption of improved maize seeds and chemical fertilizers, consequently improving their income. Results corroborate the findings by Awotide, Abdoulaye [114] that access to credit improves yields and income of cassava smallholder farmers in Nigeria. Similarly, [115] in Brazil, [20, 116] in India, and additionally [117, 118] in Kenya. Both found that farmers with access to credit can easily acquire farm implements and adopt technology and mechanized farm structures, leading to an increase in total factor productivity (TFP), translating to higher income.

## Quality of balancing

Table 8 presents the result for the quality of the matching using several key indicators to ensure adequate covariate balance observed in credit users and non-users. Before matching, the Pseudo $R^2$ was estimated at 0.033, indicating a modest explanatory power of the model. The

**Table 8. Quality of matching.**

| | PSM-NNM | | | Kernel | | | Radius | | |
|---|---|---|---|---|---|---|---|---|---|
| | Pseudo R2 | p>chi2 | Mean Bias | Pseudo R2 | p>chi2 | Mean Bias | Pseudo R2 | p>chi2 | Mean Bias |
| Unmatched | 0.033 | 0.000 | 8.9 | 0.033 | 0.000 | 8.9 | 0.033 | 0.000 | 8.9 |
| Matched | 0.007 | 0.405 | 5.1 | 0.007 | 0.447 | 4.4 | 0.007 | 0.447 | 4.6 |

likelihood-ratio test further confirmed the presence of significant differences in the distribution of covariates between the two groups (p < 0.001, indicating substantial differences in covariate distributions between the two groups suggesting that the model had strong explanatory power in distinguishing treated from control observations. The mean standardized bias was also relatively high at 8.9%, reflecting considerable imbalance. After implementing the matching procedure, the covariate balance improved significantly across all methods. Pseudo-$R^2$ values dropped to between 0.006 and 0.007, and likelihood-ratio tests became insignificant (p-values 0.447–0.494), revealing inefficiency differences in covariate alignments for microcredit users and non-users. The standardized mean difference, or Mean bias, was reduced by approximately half to 4.4%–4.8%, below the 5% threshold for adequate balance [103]. These results confirm that the matching process effectively minimized selection bias, enhancing the reliability of the impact estimates.

## Robustness

**Sensitivity analysis.** Sensitivity analysis is conducted in the final stage of non-randomized experiments, as it assesses the robustness of inferences against potential unobserved factors that could introduce bias [119]. It is crucial for solving optimization problems. As shown in Table 9, Gamma (Γ) is used in sensitivity analysis to reveal validity of the propensity score results, even when considering unobserved confounders, confirming that bias from unmeasured covariates is negligible, and all relevant covariates were included in the treatment group. Typically, treatment estimators are sensitive to unmeasured covariates that may influence causal effects due to unobserved confounders, affecting the reliability of impact evaluation [120]. To address these concerns, we illustrate the optimization problem as asymptotically separable, with bias testing resulting in the Conditional Independence Assumption (CIA). The sensitivity analysis using Rosenbaum bounds demonstrates that the positive impact of microcredit access on household income is robust to unobserved confounders [119]. The Average Treatment Effect on the Treated (ATT) estimates remained statistically significant across

**Table 9. Rosenbaum bounding sensitivity analysis results.**

| Gamma* | sig+ | sig- | t-hat+ | that- | CI+ | CI- |
|---|---|---|---|---|---|---|
| 1 | 5.1e-07 | 5.1e-07 | 11.578 | 11.578 | 11.4923 | 11.654 |
| 1.1 | .000018 | 7.7e-09 | 11.538 | 11.6181 | 11.4501 | 11.6827 |
| 1.2 | .00028 | 9.6e-11 | 11.5 | 11.6476 | 11.4245 | 11.7141 |
| 1.3 | .002336 | 1.0e-12 | 11.4672 | 11.6696 | 11.3924 | 11.7329 |
| 1.4 | .011996 | 9.3e-15 | 11.434 | 11.7001 | 11.3683 | 11.7523 |
| 1.5 | .041998 | 1.1e-16 | 11.4132 | 11.7168 | 11.3392 | 11.7758 |
| 1.6 | .108474 | 0 | 11.3902 | 11.7351 | 11.3186 | 11.8066 |
| 1.7 | .219905 | 0 | 11.3727 | 11.7488 | 11.2991 | 11.8365 |
| 1.8 | .367899 | 0 | 11.3495 | 11.7707 | 11.2783 | 11.8679 |
| 1.9 | .529446 | 0 | 11.3301 | 11.7897 | 11.2595 | 11.9071 |
| 2 | .678719 | 0 | 11.3166 | 11. 8101 | 11.2413 | 11.933 |

various matching algorithms until the sensitivity parameter (Γ) reached a threshold of 1.5, following [120]. Therefore, based on sensitivity analysis, the study results are not sensitive to substantial hidden bias for credit access participation decisions.

## Conclusion

Improving agricultural productivity through the adoption of modern technologies, particularly in food production, has garnered significant attention from global policymakers, particularly in the current climate of rising food insecurity, increasing population pressure, and the challenges posed by climate change. Given the importance of capital as a critical factor of production, access to institutional credit, as well as informal and semi-formal financial sources like microcredit, is believed to play a key role in enabling rural farm households to adopt modern agricultural technologies. Additionally, such financial access helps mitigate consumption risks, address market imperfections through crop insurance, and ultimately drive higher productivity and technical efficiency. This study investigates the effects of microcredit access on the adoption of agricultural technology and its influence on the income of maize farmers in Kenya. A multistage sampling approach was employed to select a sample of 524 respondents. The analysis was conducted using descriptive statistics, probit regression, endogenous switching regression (ESR), and sensitivity analysis techniques test robustness.

The main findings of the endogenous switching regression (ESR) model show that farmers' decisions to get credit are strongly affected by their marital status, the size of their household land, the availability of extension education, the demand for and use of inorganic fertilizers, and their membership in a cooperative. Furthermore, access to credit substantially affects maize farmers' income, with influencing factors including the source of information, land size, and availability of extension education. Most importantly, the Average Treatment Effect on the Treated (ATT) reveals that farmers with access to microcredit experience an income increase of 40.52% compared to non-users. This increase is attributed to the ease and flexibility in the acquisition of high-quality inputs, mechanizing agricultural practices, efficient use of agrochemicals, mitigating consumption risks in risk-averse households, and leveraging crop insurance. Finally, given the positive effects of microcredit from agricultural technologies, as observed in the current study and other documented literature, policy reforms to relax credit constraints for rural households are required to ensure sustainability. Such policy adjustments would facilitate broader access to financial resources, allowing farmers to overcome credit barriers and experience further spillover effects of agricultural technology adoption. Additionally, microcredit can provide the opportunity for farmers to diversify into agricultural value chains, including processing and the cultivation of high-value crops, which are often overlooked. This diversification enables households not only to increase farm productivity but also to tap into more lucrative segments of the agricultural sector, enhancing their overall economic resilience.

## Policy recommendations

While credit access facilitates technology adoption and income generation, its effectiveness is limited when considering income alone. Given that credit access is masked by several heterogeneities with different terms and conditions, farmers need to use and implement the credit purposefully for intended agricultural purposes, particularly acquiring necessary technologies that guarantee high yields that translate to income. Conversely, over time, mishandling credit allocations can result in the loss of lender trust due to defaults. Additionally, given that credit is a debt requiring repayment, maintaining technical and allocative efficiency above production frontiers is crucial for profitability.

We recommend policymakers and relevant stakeholders prioritize strategies to ease liquidity bottlenecks and provide sustainable financial solutions. This includes prioritizing savings and enhancing financial literacy by strengthening farmers' cooperatives and promoting community-based models such as rotating savings and credit associations (ROSCAs) and village savings and loans associations (VSLAs). Given these implementations, farmers' access to microcredit could reduce upfront costs and benefit from collective input subsidies and training on agricultural technologies, vital to enhancing productivity and optimal yields and guaranteeing sustainable income generations.

## Research limitations

Finally, from a methodological standpoint, although we have conducted thorough robustness checks that reinforce the validity of our findings, it is essential to emphasize that these results should be viewed as correlations due to our reliance on cross-sectional data. Future research should consider establishing a panel dataset and employing fixed-effect estimators to assess the causal relationship between credit access and technology adoption in Kenya.

## Supporting information

**S1 Table. Test for multicollinearity and heteroscedasticity.**
(DOCX)

**S1 File. Survey questionnaire.**
(DOCX)

**S2 File. Global inclusive questionnaire.**
(PDF)

## Acknowledgments

The authors would like to thank the County of Uasin Gishu and all participants for their excellent support during the survey. Sincere gratitude is also expressed to the editors and two anonymous reviewers for their valuable comments. Any remaining errors are the sole responsibility of the authors.

## Author Contributions

**Conceptualization:** Shadrack Kipkogei, John Tanui.

**Data curation:** Shadrack Kipkogei.

**Formal analysis:** Shadrack Kipkogei, Gershom Mwalupaso.

**Project administration:** Jiqin Han, John Tanui.

**Software:** Gershom Mwalupaso.

**Supervision:** Jiqin Han.

**Validation:** Robert Brenya.

**Writing – original draft:** Shadrack Kipkogei.

**Writing – review & editing:** Robert Brenya.

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
