## [Decision Letter · Decision Letter 0]

3 Oct 2024

PONE-D-24-32297The Synergistic Effects of Microcredit Access and Agricultural Technology Adoption on Maize Farmer's Income in Uasin Gishu County, KenyaPLOS ONE

Dear Dr. Kipkogei,

Thank you for submitting your manuscript to PLOS ONE. After careful consideration, we feel that it has merit but does not fully meet PLOS ONE’s publication criteria as it currently stands. Therefore, we invite you to submit a revised version of the manuscript that addresses the points raised during the review process.

We look forward to receiving your revised manuscript.

Kind regards,

Abu Hayat Md. Saiful Islam

Academic Editor

PLOS ONE

Journal Requirements:

4. Please note that your Data Availability Statement is currently missing [the repository name and/or the DOI/accession number of each dataset OR a direct link to access each database]. If your manuscript is accepted for publication, you will be asked to provide these details on a very short timeline. We therefore suggest that you provide this information now, though we will not hold up the peer review process if you are unable.

6. PLOS requires an ORCID iD for the corresponding author in Editorial Manager on papers submitted after December 6th, 2016. Please ensure that you have an ORCID iD and that it is validated in Editorial Manager. To do this, go to ‘Update my Information’ (in the upper left-hand corner of the main menu), and click on the Fetch/Validate link next to the ORCID field. This will take you to the ORCID site and allow you to create a new iD or authenticate a pre-existing iD in Editorial Manager.

7. Please amend either the title on the online submission form (via Edit Submission) or the title in the manuscript so that they are identical.

8. Please include your full ethics statement in the ‘Methods’ section of your manuscript file. In your statement, please include the full name of the IRB or ethics committee who approved or waived your study, as well as whether or not you obtained informed written or verbal consent. If consent was waived for your study, please include this information in your statement as well.

9. (a)%20present%20written%20permission%20from%20the%20copyright%20holder%20to%20publish%20these%20figures%20specifically%20under%20the%20CC%20BY%204.0%20license,%20or%20(b)%20remove%20the%20figures%20from%20your%20submission:%20%0b%0ba.%20You%20may%20seek%20permission%20from%20the%20original%20copyright%20holder%20of%20Figure%201%20to%20publish%20the%20content%20specifically%20under%20the%20CC%20BY%204.0%20license.%20We%20recommend%20that%20you%20contact%20the%20original%20copyright%20holder%20with%20the%20Content%20Permission%20Form%20(http:/journals.plos.org/plosone/s/file?id=7c09/content-permission-form.pdf)%20and%20the%20following%20text:%20%E2%80%9CI%20request%20permission%20for%20the%20open-access%20journal%20PLOS%20ONE%20to%20publish%20XXX%20under%20the%20Creative%20Commons%20Attribution%20License%20(CCAL)%20CC%20BY%204.0%20(http://creativecommons.org/licenses/by/4.0/).%20Please%20be%20aware%20that%20this%20license%20allows%20unrestricted%20use%20and%20distribution,%20even%20commercially,%20by%20third%20parties.%20Please%20reply%20and%20provide%20explicit%20written%20permission%20to%20publish%20XXX%20under%20a%20CC%20BY%20license%20and%20complete%20the%20attached%20form.%E2%80%9D%20Please%20upload%20the%20completed%20Content%20Permission%20Form%20or%20other%20proof%20of%20granted%20permissions%20as%20an%20%22Other%22%20file%20with%20your%20submission.%20In%20the%20figure%20caption%20of%20the%20copyrighted%20figure,%20please%20include%20the%20following%20text:%20%E2%80%9CReprinted%20from%20%5bref%5d%20under%20a%20CC%20BY%20license,%20with%20permission%20from%20%5bname%20of%20publisher%5d,%20original%20copyright%20%5boriginal%20copyright%20year%5d.%E2%80%9D%20%0b%0bb.%20If%20you%20are%20unable%20to%20obtain%20permission%20f">We note that Figure 1 in your submission contain map/satellite images which may be copyrighted. All PLOS content is published under the Creative Commons Attribution License (CC BY 4.0), which means that the manuscript, images, and Supporting Information files will be freely available online, and any third party is permitted to access, download, copy, distribute, and use these materials in any way, even commercially, with proper attribution. For these reasons, we cannot publish previously copyrighted maps or satellite images created using proprietary data, such as Google software (Google Maps, Street View, and Earth). For more information, see our copyright guidelines: http://journals.plos.org/plosone/s/licenses-and-copyright. We require you to either (a) present written permission from the copyright holder to publish these figures specifically under the CC BY 4.0 license, or (b) remove the figures from your submission:

a. You may seek permission from the original copyright holder of Figure 1 to publish the content specifically under the CC BY 4.0 license. We recommend that you contact the original copyright holder with the Content Permission Form (http://journals.plos.org/plosone/s/file?id=7c09/content-permission-form.pdf) and the following text: “I request permission for the open-access journal PLOS ONE to publish XXX under the Creative Commons Attribution License (CCAL) CC BY 4.0 (http://creativecommons.org/licenses/by/4.0/). Please be aware that this license allows unrestricted use and distribution, even commercially, by third parties. Please reply and provide explicit written permission to publish XXX under a CC BY license and complete the attached form.” Please upload the completed Content Permission Form or other proof of granted permissions as an "Other" file with your submission. In the figure caption of the copyrighted figure, please include the following text: “Reprinted from [ref] under a CC BY license, with permission from [name of publisher], original copyright [original copyright year].”

b. If you are unable to obtain permission from the original copyright holder to publish these figures under the CC BY 4.0 license or if the copyright holder’s requirements are incompatible with the CC BY 4.0 license, please either i) remove the figure or ii) supply a replacement figure that complies with the CC BY 4.0 license. Please check copyright information on all replacement figures and update the figure caption with source information. If applicable, please specify in the figure caption text when a figure is similar but not identical to the original image and is therefore for illustrative purposes only. The following resources for replacing copyrighted map figures may be helpful: USGS National Map Viewer (public domain): http://viewer.nationalmap.gov/viewer/ The Gateway to Astronaut Photography of Earth (public domain): http://eol.jsc.nasa.gov/sseop/clickmap/ Maps at the CIA (public domain): https://www.cia.gov/library/publications/the-world-factbook/index.html and https://www.cia.gov/library/publications/cia-maps-publications/index.html NASA Earth Observatory (public domain): http://earthobservatory.nasa.gov/ Landsat: http://landsat.visibleearth.nasa.gov/ USGS EROS (Earth Resources Observatory and Science (EROS) Center) (public domain): http://eros.usgs.gov/# Natural Earth (public domain): http://www.naturalearthdata.com/

10. Please ensure that you refer to Figure 1 in your text as, if accepted, production will need this reference to link the reader to the figure.

11. Please include captions for your Supporting Information files at the end of your manuscript, and update any in-text citations to match accordingly. Please see our Supporting Information guidelines for more information: http://journals.plos.org/plosone/s/supporting-information. 

Additional Editor Comments:

**Please read carefully the reviewers comments and made changes accordingly. I agree both the reviewers that the paper need to increase the **readability ** and discussion section need to elaborate by fitting in into the existing literatures.**

Reviewers' comments:

Reviewer's Responses to Questions

**Comments to the Author**

1. Is the manuscript technically sound, and do the data support the conclusions?

Reviewer #1: Partly

Reviewer #2: Yes

2. Has the statistical analysis been performed appropriately and rigorously? 

Reviewer #1: Yes

Reviewer #2: Yes

3. Have the authors made all data underlying the findings in their manuscript fully available?

Reviewer #1: Yes

Reviewer #2: Yes

4. Is the manuscript presented in an intelligible fashion and written in standard English?

Reviewer #1: Yes

Reviewer #2: Yes

5. Review Comments to the Author

Reviewer #1: Overall, this paper investigates the impact of microcredit access on the adoption of agricultural technologies by smallholder maize farmers in sub-Saharan Africa. It examines their access to microcredit, utilization of modern agricultural technologies, and farm income. The study reveals that land size, fertilizer use, access to extension education, and membership in cooperatives positively and significantly influence microcredit access. Furthermore, access to microcredit enhances farming practices and technical efficiency. The findings of this study have important implications for the adoption of modern technologies and poverty alleviation among farmers.

Changes are recommended in the following areas:

（1）The study does not adequately address how its findings relate to previous research in this field. The authors should revise the Introduction and Discussion sections to reference relevant literature. It is particularly important to emphasize the dialogue between this paper and existing studies in order to systematically clarify the logical relationships among microcredit access, the adoption of modern agricultural technologies, and farm income.

（2）While the study appears to be sound, the language is unclear, which makes it difficult to follow. I recommend that the authors collaborate with a writing coach or copyeditor to enhance the flow and readability of the text. In particular, the text struggles to systematically illustrate the logical relationship between access to microcredit, the use of modern agricultural technologies, and agricultural income. This lack of clarity obscures the logical connections and hinders the reader's ability to clearly understand the objectives of the various chapters.

（3）It is recommended to enhance the elaboration of the discussion section. If engaging in dialogue with similar research findings in academia proves challenging, it is essential to provide a more detailed examination of the findings presented in this paper in relation to existing literature. Improving the discussion section will strengthen the necessity and value of the research conducted in this paper.

Reviewer #2: The English writing and language should be improved, and the authors should supply the details output file of ESR model to check the consistency of the model. Which statistical analysis tool/software is used should be mentioned. I think this will be a great work if the comments will be met.

6. PLOS authors have the option to publish the peer review history of their article (what does this mean?). If published, this will include your full peer review and any attached files.

Reviewer #1: No

Reviewer #2: **Yes: **Dr. Mohammad Shamsul Hoq

---

## [Author Response · Author response to Decision Letter 0]

22 Nov 2024

Response to the Associate editor's comments.

Thank you for sharing your comments on our manuscript. We appreciate the critical feedback provided, which is invaluable for improving our work. To the best of our ability, we have tried to address the comment accordingly and red color track changes are evident in the manuscript. We hope the revised meets the standard for publication in Plos One.

Comment 1. “Please amend the title either on the online submission form or in your so that they are identical”

Response 1: We appreciate your constructive feedback regarding the title similarity, we have changed it to as it appears in reviewers with track changes.

The Synergistic Effects of Microcredit Access and Agricultural Technology Adoption on Maize Farmer's Income in Kenya

Reason for adjustment:

a) Broad Geographic relevance, all authors suggested that mentioning "Kenya" instead of a specific county (Uasin Gishu) broadens the study's appeal to an international audience. It implies that the findings may have relevance beyond the specific region.

b) Global perspective. A broader title might attract more readers, as specific regional names like Uasin Gishu may not resonate with those unfamiliar with the area.

Response to Reviewer #1

Comment 2: Abstract: The abstract is poorly written. The authors should start the abstract with a strong research problem statement. Then the aim of the study, materials and methods, findings of the study and conclusion and recommendation should be stated properly. 

Response 2: We appreciate your constructive feedback regarding the abstract. In response, we have revised the abstract by incorporating the ideal structure and content, as suggested, and have shared the updated version with all co-authors. We have ensured that the abstract clearly begins with a strong and well-defined research problem statement, aligning with your suggestion to provide a focused and direct introduction. Following this, we have outlined the aim of the study, the materials and methods employed, the key empirical findings, and concluded with a succinct summary of the recommendations. We believe the revised abstract now meets the expected structure and effectively summarizes the study’s key elements in a concise manner, in line with your guidance as shown below:

Abstract.

Addressing global food security requires urgent improvement in agricultural productivity, particularly in developing economies where market imperfections are perverse and resource constraints prevail. While microcredit is broadly recognized as a catalyst for economic empowerment, its role in facilitating agricultural technology adoption and enhancing agricultural incomes remains underexplored. This study examines the synergistic effects of microcredit access and agricultural technology adoption on the incomes of maize farmers in Kenya. Using household-level data, we employ an endogenous switching regression framework to regulate possible endogeneity in access to microcredit. We find that microcredit access positively influences the adoption of advanced agricultural technologies. Key determinants, including marital status, use of fertilizer application, access to extension services, and cooperative membership, are found to increase the probability of microcredit access significantly. Most importantly, the Average Treatment Effect on the Treated (ATT) indicates a 40.52% income increase among farmers who access credit, attributed to the timely adoption of better-quality seeds, improved agricultural technologies, and inputs. This highlights microcredit’s role in promoting allocative efficiency and enhancing Total Factor Productivity (TFP) in agricultural practices. This is attributed to the fact that robustness checks, including propensity score matching and sensitivity analyses, corroborate these findings. The study advocates for the implementation of targeted financial policies and educational initiatives to promote credit system integration, savings, and financial literacy, particularly for credit-constrained households, thereby strengthening rural financial markets and driving sustainable agricultural development across the regions.

Keywords: Microcredit, Technology Adoption, Smallholder Farmers, Endogenous Switching Regression, Total Factor Productivity, Kenya

Comment 2: The introduction explains the conceptual background and justification of a study which lead to a clear statement of research objective and scope. In introduction part, the authors describe much on microcredit aspect of different things. The authors should logically describe the issues in national and global aspect such as (1) what are the existing technologies available in case of maize farmers in Kenya? (2) what are the hinders of technology adoption? (3) what is the exiting national credit policy in the Kenya. These should be described logically.

Response 2: We appreciate your feedback regarding the abstract. In response, we shared the ideal sections of the abstract to ensure a clearer and more structured presentation with all authors as per suggestion and findings from empirical results. We have started with a stronger research problem statement which is indeed conform to your opinion as we later take keenly on importance of beginning with proper and strong straightforward statement, followed by the aim of the study, materials and methods used, key findings, and concluded with a clear summary of the recommendations. We believe the revised abstract now adheres to the expected structure and concisely summarizes the study.

2. Introduction: We appreciate your valuable feedback on the introduction. In response to your suggestions, we have made substantial revisions to this section to ensure it more clearly addresses the national and global context, as well as the rationale for the study as illustrated in paragraph 2-4 page 1 with each trying to answer the following after reading and incorporating updated data. 

National and Global Context: We have restructured the introduction to logically present the issues related to maize farming in Kenya within both national and global perspectives. Specifically, we now highlight: Existing Technologies for Maize Farmers in Kenya: A comprehensive review of the available technologies has been included to contextualize the current practices and innovations in maize farming in Kenya.

a) Modern technologies existing in Kenya maize farmers include improved seed varieties, inorganic fertilizers, mechanization, and irrigation. However, adoption of these technologies remains low among smallholder farmers, who based on resource constraints rely on traditional practices (1). Fertilizer usage, for instance, averages 20 kg/ha far below global benchmarks in regions such as China, Brazil, India, and South Africa. This indicates a critical gap in nutrient input that hampers crop yields (FAO, 2022). Similarly, mechanization in Kenya stands at just 25%, implying that most farmers still depend on manual labor in various production stages which is less efficient and constrains productivity (2). Moreover, only 7% of Kenya’s agricultural land is irrigated, leaving most farmers to rely on unpredictable rainfall patterns (3). This insufficient utilization of available technologies reveals the urgency of improving credit accessibility, strengthening extension services, and upgrading infrastructure to promote the adoption of modern farming technologies

b) Barriers to Technology Adoption: We have elaborated on the various hindrances that prevent farmers from adopting these technologies, including economic, social, and infrastructural challenges.

In Kenya, despite agricultural potential, smallholder farmers face several barriers to the adoption of modern farming technologies. High input costs particularly for fertilizers, improved seeds, and mechanization tools, make these technologies inaccessible to many farmers. Moreover, financial constraints remain a significant challenge, given that about 90% of rural farmers are unable to access formal credit, restricting their investment capacity (3). Additionally, inadequate rural infrastructure and weak agricultural extension services contribute to a lack of awareness and education on the benefits of new technologies. Furthermore, cultural preferences and risk aversion lead farmers to stick with traditional methods, fearing the risks associated with adopting unfamiliar practices. These factors collectively limit technological adoption in the agricultural sector.

c) National Credit Policies in Kenya: A thorough discussion of Kenya’s national credit policies and their impact on farmers' access to financing for technological adoption has been added. This provides a clearer understanding of the institutional. To address financial constraints, undeniably, the Kenyan government, has introduced various credit policies and programs aimed at supporting smallholder farmers. These include the Agricultural Finance Corporation (AFC), established in 1963 to provide credit for agricultural production, and Kilimo Biashara, launched in 2008 to offer affordable loans to small-scale farmers. The Uwezo Fund, initiated in 2013, targets youth empowerment, women, and people with disabilities in agribusiness. Additionally, in a quest to reduce the unemployment crisis, the Enable Youth Kenya Program was launched in 2016 with funding from the African Development Bank (AfDB). Despite these efforts, credit access remains constrained, especially in rural areas hindered by supply-side factors such as insufficient credit sources, unsuitable loan products, and lengthy procedures. Nevertheless, relaxation of supply-side factors, by lowering interest rates and increasing microfinance access in rural areas, have remained challenging as farmers may decline to use credit due to (i) asymmetric information, (ii) risk aversion, strict lending requirements, availability of alternative financing, (iii) cumbersome loan processes. Furthermore, non-contingency terms in credit contracts can limit microcredit demand, often leading to defaults and the seizure of household properties.

Research Gap and Rationale: We have refined the justification for the study by conducting a more rigorous review of the literature by covering all related studies particularly those relating to credit and technology adoption between 2015-2024 and this has helped us to delve deeper clarity identifying a gap in current research. This gap lies in the lack of comprehensive studies that explore the intersection of technology adoption, credit access, and agricultural development for maize farmers in Kenya. We address this gap after reading and making relevance with other existing works and finally we address as follow;

This study contributes to the literature in two folds. First, previous empirical studies have explored how access to credit can enhance technology adoption (4-8), as well as yields and productivity (9-11). However, they often focus on single technology packages and with no comprehensive consideration of income effects differentials between credit-constrained and non-credit-constrained households. Moreover, previous studies in Kenya have primarily examined the determinants of credit access technological adoption, focusing on mobile banking (12-14), identifying factors such as the occupation of the household head, access to financial training, and household credit liquidity constraints. However, there is a paucity of studies that assess the impact of credit access on technology adoption in a more unified manner, mainly concerning its effects on household income. Secondly, to the best of our knowledge, while much of the existing literature has focused on the direct effects of credit access on technology adoption, few studies have addressed the heterogeneous impacts of credit access across different agricultural yields and regionals (Girma and Research (15), Makate, Makate (16), Mariyono (17). This study aims to fill this gap by incorporating an endogenous switching regression (ESR) in examining how credit access influences technology adoption and income outcomes across different agricultural yields, considering that ESR addresses the potential existence of endogeneity and we provide insights on policies, in augmenting and strengthen rural financial institutional partnerships within farming communities, using credit for economic empowerment, mainly by investing in technological adoption and considering flexible repayment mechanisms.

Response 3: We have revised the discussion to ensure that it is more directly linked to the results, with a clear focus on interpreting and validating our findings. Each key result is now discussed in detail, with an emphasis on how it contributes to the overall understanding of the study as shown on first line of line 2-5 on page 14-16

The results in Table 3 summarize the social economic characteristics of the maize farmers illustrating variables used in the model estimation. The dependent variable, farmers' level of technology adoption farmers using microcredit provides perceived insights into demographic and economic adoption patterns. The average age of maize farmers is 45 years, relatively younger than 46 years for non-users, indicating that farmers are relatively young, with productive capabilities of adopting technological advancement that could enhance productivity. Results corroborate empirical findings by Bakare, Ogunleye (18) and Gabriel and Gandorfer (19) who found that the age of 41-50 is a critical point of productivity and receptivity to innovation, young farmers possess the skills and risk tolerance to incorporate new agricultural technologies effectively. Additionally, findings correspond to findings by Asante-Addo, Mockshell (20) in Ghana who found that younger farmers exhibit a proactive stance toward technology adoption, distinctive attributes caused by familiarity with digital tools, and a heightened propensity for risk-taking. In contrast, as farmers age, the adoption likelihood tends to diminish, as older farmers often prioritize stability over potential yield-enhancing investments due to increased conservatism and reliance on accumulated savings rather than external financing,(21). This pattern resonates with microeconomic theories of diminishing marginal utility in risk aversion, where the marginal benefit of new technology does not adequately outweigh the perceived risks among older farmers, Sunding and Zilberman (22)

Comparison with Existing Literature: We have expanded the discussion by referencing the relevant literature, highlighting where our findings align with or differ from previous studies. This helps to contextualize our results within the broader body of research particularly on discussion part on determinants and impacts of credit access by changing the table variables particularly in consideration of instrumented variable, we drop variable bank access as it was more correlated with other variables after testing the VIF. This has been extensively done and illustrated in last paragraph of page 17.

Further, both groups have adopted over 80% improved maize seed varieties, but credit users show higher adoption rates of inorganic fertilizers (89%) in comparison to noncredit users (19%) implying that access to these inputs is seemingly relatively expensive though crucial for optimal yields. Noncredit users generally have fewer than ten years of schooling, less involvement in financial institutions (52% have bank accounts versus 89% for credit users), and lower access to extension education and information. Distance to financial institutions and markets varies significantly. Credit users have shorter distances to markets (0.16 km) and financial institutions (1.98 km), whereas noncredit users face longer distances to markets (0.91 km) and financial institutions (2.00 km). Given that Table 3 does not provide differences in determinants and income impacts of credit access, a more robust approach is necessary; otherwise, an inexperienced estimator is likely to either under/or overstate the findings. Further, we tested for multicollinearity and heteroscedasticity as illustrated in Appendix Table 1A. According to Gujarati (23), VIF more than the threshold value of 10 indicates multicollinearity within covariates. Based on our results, as illustrated in Table

---

## [Editor Report · Decision Letter 1]

27 Nov 2024

PONE-D-24-32297R1The Synergistic Effects of Microcredit Access and Agricultural Technology Adoption on Maize Farmer's Income in KenyaPLOS ONE

Dear Dr. Kipkogei,

Thank you for submitting your manuscript to PLOS ONE. After careful consideration, we feel that it has merit but does not fully meet PLOS ONE’s publication criteria as it currently stands. Therefore, we invite you to submit a revised version of the manuscript that addresses the points raised during the review process.

We look forward to receiving your revised manuscript.

Kind regards,

Abu Hayat Md. Saiful Islam

Academic Editor

PLOS ONE
---

## [Author Response · Author response to Decision Letter 1]

3 Dec 2024

Response to the Associate editor's comments.

Thank you for sharing your comments on our manuscript. We appreciate the critical feedback provided, which is invaluable for improving our work. To the best of our ability, we have tried to address the comment accordingly and red color track changes are evident in the manuscript. We hope the revised meets the standard for publication in Plos One.

Comment 1. “Please amend the title either on the online submission form or in your so that they are identical”

Response 1: We appreciate your constructive feedback regarding the title similarity, we have changed it to as it appears in reviewers with track changes.

The Synergistic Effects of Microcredit Access and Agricultural Technology Adoption on Maize Farmer's Income in Kenya

Reason for adjustment:

a) Broad Geographic relevance, all authors suggested that mentioning "Kenya" instead of a specific county (Uasin Gishu) broadens the study's appeal to an international audience. It implies that the findings may have relevance beyond the specific region.

b) Global perspective. A broader title might attract more readers, as specific regional names like Uasin Gishu may not resonate with those unfamiliar with the area.

Response to Reviewer #1

Comment 2: Abstract: The abstract is poorly written. The authors should start the abstract with a strong research problem statement. Then the aim of the study, materials and methods, findings of the study and conclusion and recommendation should be stated properly. 

Response 2: We appreciate your constructive feedback regarding the abstract. In response, we have revised the abstract by incorporating the ideal structure and content, as suggested, and have shared the updated version with all co-authors. We have ensured that the abstract clearly begins with a strong and well-defined research problem statement, aligning with your suggestion to provide a focused and direct introduction. Following this, we have outlined the aim of the study, the materials and methods employed, the key empirical findings, and concluded with a succinct summary of the recommendations. We believe the revised abstract now meets the expected structure and effectively summarizes the study’s key elements in a concise manner, in line with your guidance as shown below:

Abstract.

Addressing global food security requires urgent improvement in agricultural productivity, particularly in developing economies where market imperfections are perverse and resource constraints prevail. While microcredit is broadly recognized as a catalyst for economic empowerment, its role in facilitating agricultural technology adoption and enhancing agricultural incomes remains underexplored. This study examines the synergistic effects of microcredit access and agricultural technology adoption on the incomes of maize farmers in Kenya. Using household-level data, we employ an endogenous switching regression framework to regulate possible endogeneity in access to microcredit. We find that microcredit access positively influences the adoption of advanced agricultural technologies. Key determinants, including marital status, use of fertilizer application, access to extension services, and cooperative membership, are found to increase the probability of microcredit access significantly. Most importantly, the Average Treatment Effect on the Treated (ATT) indicates a 40.52% income increase among farmers who access credit, attributed to the timely adoption of better-quality seeds, improved agricultural technologies, and inputs. This highlights microcredit’s role in promoting allocative efficiency and enhancing Total Factor Productivity (TFP) in agricultural practices. This is attributed to the fact that robustness checks, including propensity score matching and sensitivity analyses, corroborate these findings. The study advocates for the implementation of targeted financial policies and educational initiatives to promote credit system integration, savings, and financial literacy, particularly for credit-constrained households, thereby strengthening rural financial markets and driving sustainable agricultural development across the regions.

Keywords: Microcredit, Technology Adoption, Smallholder Farmers, Endogenous Switching Regression, Total Factor Productivity, Kenya

Comment 2: The introduction explains the conceptual background and justification of a study which lead to a clear statement of research objective and scope. In introduction part, the authors describe much on microcredit aspect of different things. The authors should logically describe the issues in national and global aspect such as (1) what are the existing technologies available in case of maize farmers in Kenya? (2) what are the hinders of technology adoption? (3) what is the exiting national credit policy in the Kenya. These should be described logically.

Response 2: We appreciate your feedback regarding the abstract. In response, we shared the ideal sections of the abstract to ensure a clearer and more structured presentation with all authors as per suggestion and findings from empirical results. We have started with a stronger research problem statement which is indeed conform to your opinion as we later take keenly on importance of beginning with proper and strong straightforward statement, followed by the aim of the study, materials and methods used, key findings, and concluded with a clear summary of the recommendations. We believe the revised abstract now adheres to the expected structure and concisely summarizes the study.

2. Introduction: We appreciate your valuable feedback on the introduction. In response to your suggestions, we have made substantial revisions to this section to ensure it more clearly addresses the national and global context, as well as the rationale for the study as illustrated in paragraph 2-4 page 1 with each trying to answer the following after reading and incorporating updated data. 

National and Global Context: We have restructured the introduction to logically present the issues related to maize farming in Kenya within both national and global perspectives. Specifically, we now highlight: Existing Technologies for Maize Farmers in Kenya: A comprehensive review of the available technologies has been included to contextualize the current practices and innovations in maize farming in Kenya.

a) Modern technologies existing in Kenya maize farmers include improved seed varieties, inorganic fertilizers, mechanization, and irrigation. However, adoption of these technologies remains low among smallholder farmers, who based on resource constraints rely on traditional practices (1). Fertilizer usage, for instance, averages 20 kg/ha far below global benchmarks in regions such as China, Brazil, India, and South Africa. This indicates a critical gap in nutrient input that hampers crop yields (FAO, 2022). Similarly, mechanization in Kenya stands at just 25%, implying that most farmers still depend on manual labor in various production stages which is less efficient and constrains productivity (2). Moreover, only 7% of Kenya’s agricultural land is irrigated, leaving most farmers to rely on unpredictable rainfall patterns (3). This insufficient utilization of available technologies reveals the urgency of improving credit accessibility, strengthening extension services, and upgrading infrastructure to promote the adoption of modern farming technologies

b) Barriers to Technology Adoption: We have elaborated on the various hindrances that prevent farmers from adopting these technologies, including economic, social, and infrastructural challenges.

In Kenya, despite agricultural potential, smallholder farmers face several barriers to the adoption of modern farming technologies. High input costs particularly for fertilizers, improved seeds, and mechanization tools, make these technologies inaccessible to many farmers. Moreover, financial constraints remain a significant challenge, given that about 90% of rural farmers are unable to access formal credit, restricting their investment capacity (3). Additionally, inadequate rural infrastructure and weak agricultural extension services contribute to a lack of awareness and education on the benefits of new technologies. Furthermore, cultural preferences and risk aversion lead farmers to stick with traditional methods, fearing the risks associated with adopting unfamiliar practices. These factors collectively limit technological adoption in the agricultural sector.

c) National Credit Policies in Kenya: A thorough discussion of Kenya’s national credit policies and their impact on farmers' access to financing for technological adoption has been added. This provides a clearer understanding of the institutional. To address financial constraints, undeniably, the Kenyan government, has introduced various credit policies and programs aimed at supporting smallholder farmers. These include the Agricultural Finance Corporation (AFC), established in 1963 to provide credit for agricultural production, and Kilimo Biashara, launched in 2008 to offer affordable loans to small-scale farmers. The Uwezo Fund, initiated in 2013, targets youth empowerment, women, and people with disabilities in agribusiness. Additionally, in a quest to reduce the unemployment crisis, the Enable Youth Kenya Program was launched in 2016 with funding from the African Development Bank (AfDB). Despite these efforts, credit access remains constrained, especially in rural areas hindered by supply-side factors such as insufficient credit sources, unsuitable loan products, and lengthy procedures. Nevertheless, relaxation of supply-side factors, by lowering interest rates and increasing microfinance access in rural areas, have remained challenging as farmers may decline to use credit due to (i) asymmetric information, (ii) risk aversion, strict lending requirements, availability of alternative financing, (iii) cumbersome loan processes. Furthermore, non-contingency terms in credit contracts can limit microcredit demand, often leading to defaults and the seizure of household properties.

Research Gap and Rationale: We have refined the justification for the study by conducting a more rigorous review of the literature by covering all related studies particularly those relating to credit and technology adoption between 2015-2024 and this has helped us to delve deeper clarity identifying a gap in current research. This gap lies in the lack of comprehensive studies that explore the intersection of technology adoption, credit access, and agricultural development for maize farmers in Kenya. We address this gap after reading and making relevance with other existing works and finally we address as follow;

This study contributes to the literature in two folds. First, previous empirical studies have explored how access to credit can enhance technology adoption (4-8), as well as yields and productivity (9-11). However, they often focus on single technology packages and with no comprehensive consideration of income effects differentials between credit-constrained and non-credit-constrained households. Moreover, previous studies in Kenya have primarily examined the determinants of credit access technological adoption, focusing on mobile banking (12-14), identifying factors such as the occupation of the household head, access to financial training, and household credit liquidity constraints. However, there is a paucity of studies that assess the impact of credit access on technology adoption in a more unified manner, mainly concerning its effects on household income. Secondly, to the best of our knowledge, while much of the existing literature has focused on the direct effects of credit access on technology adoption, few studies have addressed the heterogeneous impacts of credit access across different agricultural yields and regionals (Girma and Research (15), Makate, Makate (16), Mariyono (17). This study aims to fill this gap by incorporating an endogenous switching regression (ESR) in examining how credit access influences technology adoption and income outcomes across different agricultural yields, considering that ESR addresses the potential existence of endogeneity and we provide insights on policies, in augmenting and strengthen rural financial institutional partnerships within farming communities, using credit for economic empowerment, mainly by investing in technological adoption and considering flexible repayment mechanisms.

Response 3: We have revised the discussion to ensure that it is more directly linked to the results, with a clear focus on interpreting and validating our findings. Each key result is now discussed in detail, with an emphasis on how it contributes to the overall understanding of the study as shown on first line of line 2-5 on page 14-16

The results in Table 3 summarize the social economic characteristics of the maize farmers illustrating variables used in the model estimation. The dependent variable, farmers' level of technology adoption farmers using microcredit provides perceived insights into demographic and economic adoption patterns. The average age of maize farmers is 45 years, relatively younger than 46 years for non-users, indicating that farmers are relatively young, with productive capabilities of adopting technological advancement that could enhance productivity. Results corroborate empirical findings by Bakare, Ogunleye (18) and Gabriel and Gandorfer (19) who found that the age of 41-50 is a critical point of productivity and receptivity to innovation, young farmers possess the skills and risk tolerance to incorporate new agricultural technologies effectively. Additionally, findings correspond to findings by Asante-Addo, Mockshell (20) in Ghana who found that younger farmers exhibit a proactive stance toward technology adoption, distinctive attributes caused by familiarity with digital tools, and a heightened propensity for risk-taking. In contrast, as farmers age, the adoption likelihood tends to diminish, as older farmers often prioritize stability over potential yield-enhancing investments due to increased conservatism and reliance on accumulated savings rather than external financing,(21). This pattern resonates with microeconomic theories of diminishing marginal utility in risk aversion, where the marginal benefit of new technology does not adequately outweigh the perceived risks among older farmers, Sunding and Zilberman (22)

Comparison with Existing Literature: We have expanded the discussion by referencing the relevant literature, highlighting where our findings align with or differ from previous studies. This helps to contextualize our results within the broader body of research particularly on discussion part on determinants and impacts of credit access by changing the table variables particularly in consideration of instrumented variable, we drop variable bank access as it was more correlated with other variables after testing the VIF. This has been extensively done and illustrated in last paragraph of page 17.

Further, both groups have adopted over 80% improved maize seed varieties, but credit users show higher adoption rates of inorganic fertilizers (89%) in comparison to noncredit users (19%) implying that access to these inputs is seemingly relatively expensive though crucial for optimal yields. Noncredit users generally have fewer than ten years of schooling, less involvement in financial institutions (52% have bank accounts versus 89% for credit users), and lower access to extension education and information. Distance to financial institutions and markets varies significantly. Credit users have shorter distances to markets (0.16 km) and financial institutions (1.98 km), whereas noncredit users face longer distances to markets (0.91 km) and financial institutions (2.00 km). Given that Table 3 does not provide differences in determinants and income impacts of credit access, a more robust approach is necessary; otherwise, an inexperienced estimator is likely to either under/or overstate the findings. Further, we tested for multicollinearity and heteroscedasticity as illustrated in Appendix Table 1A. According to Gujarati (23), VIF more than the threshold value of 10 indicates multicollinearity within covariates. Based on our results, as illustrated in Table

---

## [Editor Report · Decision Letter 2]

4 Dec 2024

The Synergistic Effects of Microcredit Access and Agricultural Technology Adoption on Maize Farmer's Income in Kenya

PONE-D-24-32297R2

Dear Dr. Shadrack Kipkogei,

We’re pleased to inform you that your manuscript has been judged scientifically suitable for publication and will be formally accepted for publication once it meets all outstanding technical requirements.

Kind regards,

Abu Hayat Md. Saiful Islam

Academic Editor

PLOS ONE

Additional Editor Comments (optional):

Please follow the PLOS ONE guidelines for formatting and referencing as well as please check the typos during final revision. 
---

## [Editor Report · Acceptance letter]

26 Dec 2024

PONE-D-24-32297R2 

PLOS ONE

Dear Dr. Han, 

I'm pleased to inform you that your manuscript has been deemed suitable for publication in PLOS ONE. Congratulations! Your manuscript is now being handed over to our production team.

Kind regards, 

on behalf of

Dr. Abu Hayat Md. Saiful Islam 

Academic Editor

PLOS ONE